# Adhesion energy controls lipid binding-mediated endocytosis

Raluca Groza[1], Kita Valerie Schmidt[1,2], Paul Markus Müller [1], Paolo Ronchi [3], Claire Schlack-Leigers[1], Ursula Neu[1], Dmytro Puchkov [4], Rumiana Dimova [2], Claudia Matthaeus[5,6], Justin Taraska [5], Thomas R. Weikl [2] & Helge Ewers [1] ✉

Several bacterial toxins and viruses can deform membranes through multivalent binding to lipids for clathrin-independent endocytosis. However, it remains unclear, how membrane deformation and endocytic internalization are mechanistically linked. Here we show that many lipid-binding virions induce membrane deformation and clathrin-independent endocytosis, suggesting a common mechanism based on multivalent lipid binding by globular particles. We create a synthetic cellular system consisting of a lipid-anchored receptor in the form of GPI-anchored anti-GFP nanobodies and a multivalent globular binder exposing 180 regularly-spaced GFP molecules on its surface. We show that these globular, 40 nm diameter, particles bind to cells expressing the receptor, deform the plasma membrane upon adhesion and become endocytosed in a clathrin-independent manner. We explore the role of the membrane adhesion energy in endocytosis by using receptors with affinities varying over 7 orders of magnitude. Using this system, we find that once a threshold in adhesion energy is overcome to allow for membrane deformation, endocytosis occurs reliably. Multivalent, binding-induced membrane deformation by globular binders is thus sufficient for internalization to occur and we suggest it is the common, purely biophysical mechanism for lipid-binding mediated endocytosis of toxins and pathogens.

Endocytosis can occur either through a clathrin-mediated process (CME)[1,2] or in a clathrin-independent manner (CIE)[3,4]. Regardless of the mechanism, the first step involves the deformation of the plasma membrane into a nanoscale invagination that buds into a vesicle. For this to occur, the stiffness of the membrane must be overcome[5,6]. In CME, the energy required to deform the membrane into an endocytic pit stems from active and passive processes in a complex sequence of events. The clathrin-coated pit that is formed in this process stabilizes the budding vesicle by the consumption of energy released by clathrin assembly[7–13]. Clathrin-independent endocytosis by definition does not

include this process and many molecules internalized by such pathways are capable of deforming membranes by themselves[14–16]. Indeed several cargoes that bind multivalently to glycolipids such as Galectin3[17], the Cholera toxin beta subunit (CTxB)[15,18], the Shiga toxin[14] and Simian Virus 40 (SV40)[15] can deform membranes in vitro and in energy-depleted cells. This membrane deformation seems to be dependent on the presence of several binding sites in close proximity, since antibodies to glycolipids[15] or a CTxB mutant, in which all but one binding sites are mutated[19,20], cannot deform membranes. Furthermore, globular glycolipid binders such as SV40 can deform even stiff

[1]Institute of Biochemistry, Freie Universität Berlin, Thielallee 63, 14195 Berlin, Germany. [2]Max Planck Institute of Colloids and Interfaces, Potsdam Science Park, Am Mühlenberg 1, 14476 Potsdam, Germany. [3]Electron Microscopy Core Facility, European Molecular Biology Laboratory, 69117 Heidelberg, Germany. [4]Leibniz-Forschungsinstitut für Molekulare Pharmakologie (FMP), 13125 Berlin, Germany. [5]National Heart Lung and Blood Institute, National Institutes of Health, Bethesda, MD 20892, USA. [6]Present address: Institute for Nutritional Science, University of Potsdam, Arthur-Scheunert-Allee 114-116, 14558 Nuthetal, Germany. ✉e-mail: helge.ewers@fu-berlin.de

membranes, that CTxB cannot deform, by imprinting their ~ 45 nm diameter shape on the plasma membrane of host cells[15]. This suggests that an interplay between particle shape and adhesion energy applied from several binding sites in a nanoscopic domain may provide a common biophysical mechanism for membrane deformation and endocytosis. However, no tractable experimental system to test the role of adhesion energy in such a model is available.

Here, we reconstitute an artificial ligand-receptor system that consists of a 40 nm diameter, polyvalent globular binder and lipidic receptors in cells. Specifically, we use a polymerized capsid studded with 180 GFP molecules and anti-GFP nanobodies on GPI-anchors as receptors. In our system, particles deform cellular membranes and become endocytosed in a clathrin-independent manner. When we modified the adhesion energy forced upon membranes by particles by changing receptor affinity from 36 pM to 23 mM, we found that a specific adhesion energy threshold was required for membrane deformation and endocytosis. Our findings provide a simple mechanism that may explain how many bacterial toxins and viruses exploit multivalent lipid binding for internalization.

## Results

### Membrane deformation after polyvalent lipid binding is a common mechanism of viral endocytosis

Our initial objective was to investigate whether a common mechanism for membrane deformation and endocytosis for globular multivalent lipid-binding nanoscale particles exists. To do so, we made use of virus-like particles assembled from non-enveloped, ~45 nm diameter lipid-binding viruses. Specifically, we generated VLPs of Simian Virus 40 (SV40), murine polyomavirus (mPy) and JC Virus (JCV), which bind to the gangliosides GM1 (SV40[21]), GD1a or GT1b (mPy[21–23]) and GD1b (JCV[24]), respectively. Particles were assembled from the respective ganglioside-binding coat protein VP1 only (Supplementary Fig. S1A) and fluorescence labeled VLPs bound readily and in a monodisperse manner to cells (Supplementary Fig. S1B). When we added fluorescence-labeled VLPs to giant unilamellar vesicles (GUVs) containing the respective receptor ganglioside, we found that all particles deformed the membrane bilayer into tubular structures emanating into the lumen of the GUV (Fig. 1A and Supplementary Fig. S1C). To observe membrane deformation in cells, we depleted cells of energy by treatment with deoxy-glucose and sodium azide to disrupt active processes that lead to scission and added VLPs to them, we found that when bound to cellular plasma membranes, VLPs likewise deformed them. Membrane-bound VLPs induced the formation of tubular invaginations that were continuous with the plasma membrane and extended into the cytosol for VLPs from all viruses (Fig. 1B). We concluded that membrane deformation is a common feature for globular glycolipid-binding multivalent ligands. We next asked, whether VLPs would become endocytosed. Our results demonstrate that they readily internalize (Supplementary Fig. S1D,E) and accumulate in LAMP-1 positive endosomal structures inside cells as shown before for SV40 (ref. 25 and Fig. 1D). Thus, the multivalent globular lipid-binding VLPs are all capable of deforming artificial and cellular membranes and became internalized and trafficked through the endolysosomal system, suggesting a common underlying principle based on lipid binding.

### A polyvalent virus-like-particle-lipid receptor system for endocytosis

We next aimed to determine if multivalent lipid binding alone may suffice for particles to become endocytosed. To do so, we generated a synthetic particle-receptor system using a genetically encoded nanoparticle (GEM[26]) bearing 180 copies of GFP on its surface. The GEMs self-assembled from monomers of the *encapsulin* protein from the archaeon *Pyrococcus furiosus*[27] coupled to GFP when recombinantly expressed in *E.coli* and formed ~40 nm diameter globular particles

(Fig. 2A). We refer to these particles from now on as GEMs and employ them as model for a nanoscale globular lipid binder. As receptor we used anti-GFP nanobodies[28] incorporated into the plasma membrane via a glycosylphosphatidylinositol anchor (GPI-anchor) upon transient expression (Fig. 2A). Purified GEMs appeared as ~40 nm diameter monodisperse particles in transmission electron microscopy and when added to cells expressing the receptor, bound as discrete fluorescent spots exhibiting lateral motion (Fig. 2B and Supplementary Movie 1). Binding was specific and could be blocked by the addition of free nanobodies to the medium (Supplementary Fig. S2A). When we generated Giant Plasma Membrane Vesicles (GPMVs) from receptor-expressing cells, GEMs bound to them readily and, after a period of adhering to and accumulating on the membrane, formed tubular invaginations emanating into the lumen of the GPMVs (Fig. 2C).

Moreover, when we depleted energy of the receptor-expressing cells using sodium azide/deoxy-glucose as above, the GEMs were observed to induce membrane curvature in a similar manner to the deformation we observed for glycolipid-binding virions (Fig. 2D). We concluded that GEMs could induce membrane deformation in plasma membranes of cells. To probe whether membrane-bound GEMs colocalized with known endocytic structures such as clathrin or caveolae, we added GEMs to live cells expressing clathrin-light chain-DsRed or caveolin1-mRFP and observed them in total internal reflection fluorescence (TIRF) microscopy. We found that membrane-bound GEMs were virtually absent from clathrin-DsRed positive areas and merely a small fraction colocalized with caveolin1-mRFP (Fig. 2E, F and Supplementary Fig. S7). To identify a possible early endocytic structure, we then performed correlative platinum-replica electron microscopy and confocal fluorescence microscopy of unroofed cells[29,30] after GEM binding (Fig. 2G, Supplementary Fig. S2B, C and Supplementary Movie 2). We found that areas with bound GEMs exhibited small, irregularly shaped membrane structures that were mostly devoid of the characteristic clathrin-coats or caveolar ribbon structures (Fig. 2G,H and Supplementary Fig. S2B). The green correlative fluorescence is clearly elongated and thus not resulting from a single diffraction-limited spot. It must thus be the result of several particles located in an elongated arrangement like in a tubule (Fig. 2G). We concluded that the GEMs were capable of deforming membranes into small invaginations emanating from the plasma membrane into the cytosol in live cells independently of both clathrin and caveolae.

When we imaged GEMs bound to cells for longer periods in live-cell microscopy, we observed that they became trapped in bright structures that moved in a directed manner inside cells, suggesting that GEMs became endocytosed into vesicular structures containing many particles. To investigate the identity of these endocytic carrier vesicles, we employed Correlative Light Electron Microscopy (CLEM) to precisely resolve the intracellular localization of the particles. To do so, we bound 5 µg/ml of GEMs to cells and incubated them at 37 °C to allow for internalization and intracellular transport. We then performed high-pressure-freezing on these cells after 1 h or 6 h incubation and performed cryogenic CLEM. When we acquired low magnification correlative light-electron microscopy overviews of these cells (Supplementary Fig. S3A), we were able to pinpoint the precise localization of the GEMs inside the cells using their fluorescence signal. High-magnification transmission electron-tomograms of the intracellular compartments positive for GFP signal revealed internalized GEMs as small dark particles of about 50 nm diameter (Fig. 3A, B and Supplementary Movie 3). GEMs were located to structures we could morphologically identify to be either endosomes or lysosomes for both time points (Fig. 3A, B, Supplementary Fig. S3B). These results were consistent with observations we made by live-cell microscopy on a spinning-disk confocal microscope where the GFP fluorescence of the GEMs increasingly colocalized with that of Rab7-mRFP and Lamp1-mRFP in perinuclear vesicular structures over several hours of endocytosis (Fig. 3C, D and Supplementary Fig. S3C). To test by what

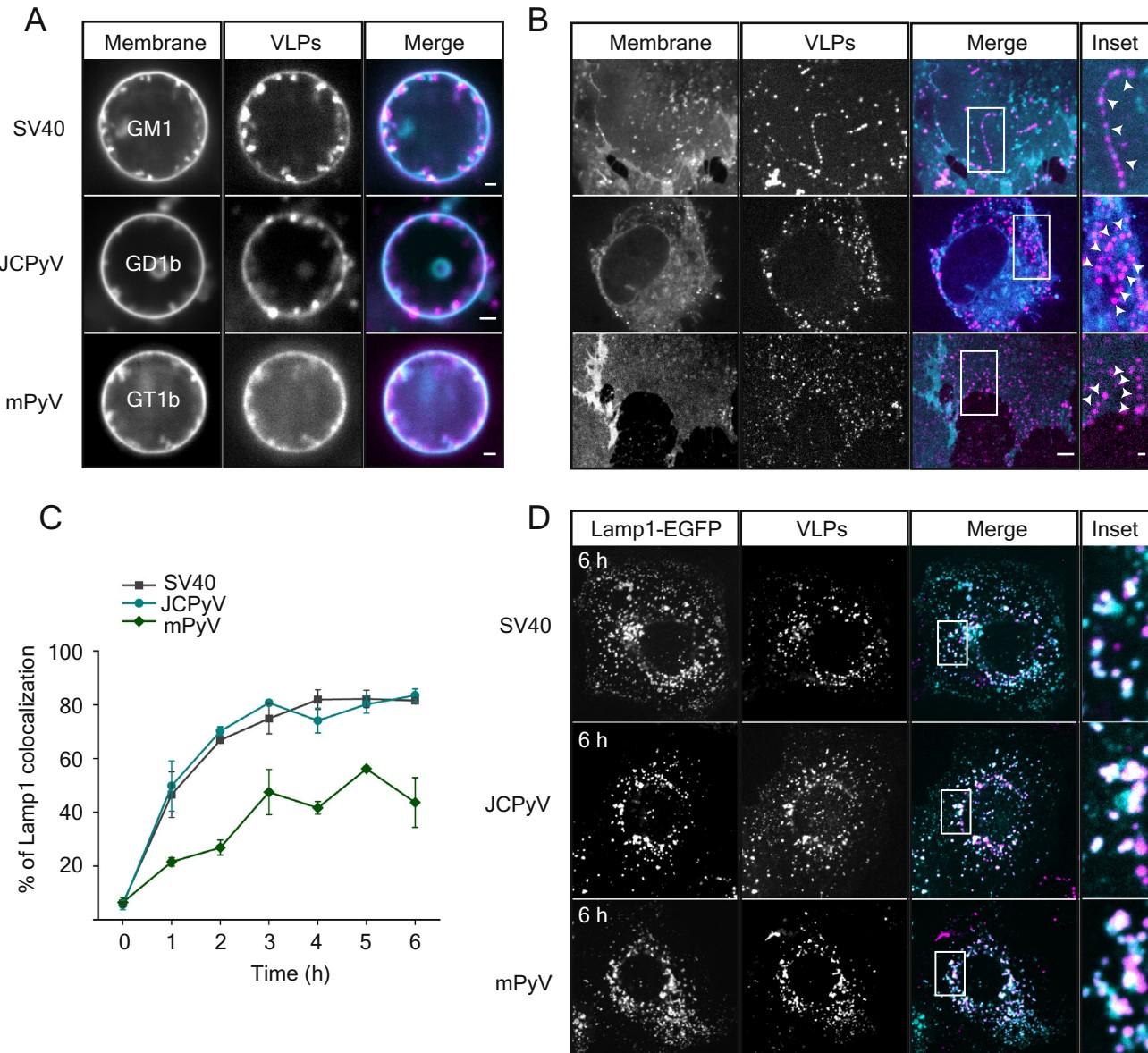

**Fig. 1 | Membrane deformation after polyvalent lipid binding is a common mechanism of viral endocytosis. A** Spinning disc confocal fluorescence microscopy micrographs of polyomavirus-like particles (VLPs) bound to Giant unilamellar vesicles (GUVs) containing receptor gangliosides. 2 µg of each VLP was incubated for 1 h at RT with GUVs containing the indicated gangliosides (98% DOPC, 1% ganglioside, 1% β-BODIPY FL C12-HPC dye) and imaged at the equatorial plane. Scale bar is 2 µm. **B** Spinning disc confocal fluorescence microscopy micrographs of polyoma VLPs bound to energy-depleted CV1 cells. Cells were starved of cellular energy by 30 min incubation in starvation buffer (PBS +/+ supplemented with 10 mM 2-deoxy-D-glucose and 10 mM NaN3) followed by 1 h incubation with 5 µg of each VLP in starvation buffer and imaged live on a spinning disk confocal microscope. DiI membrane dye was added 10 min prior to imaging at

1 mg/ml final concentration. Scale bars are 5 µm and 1 µm for insets. Arrows mark VLP-filled membrane invaginations. **C** Quantification of colocalization in confocal fluorescence micrographs between polyomavirus VLPs and lysosomes as marked by Lamp1-GFP in live cells. CV1 cells expressing Lamp1-EGFP were kept at 4 °C for 10 min before incubation with 2 µg of the indicated VLPs for 30 min at 4 °C, followed by further incubation at 37 °C for the indicated times before imaging live on a spinning disk confocal microscope. Means ± s.e.m. from $n = 3$ independent experiments. **D** Representative confocal fluorescence micrographs of the Lamp1-EGFP expressing cells containing the indicated VLPs after 6 h incubation at 37 °C. Scale bars are 5 µm and 1 µm for insets. Magenta: VLPs, Cyan: membrane marker or Lamp1-EGFP. Source data are provided as a Source Data file.

specific mechanism GEMs became endocytosed, we developed a fluorescence-assisted cell sorting (FACS) based internalization assay (Supplementary Fig. S4A). To do so, we added 2 µg/ml of GEMs to cells and incubated them at 37 °C for 1 h. We then washed cells in acidic buffer, which efficiently removed surface-bound GEMS, but not internalized particles (Supplementary Fig. S4B), allowing us to quantify endocytosis. We then used our FACS-based internalization assay with cells transfected with siRNA against clathrin-heavy chain (CHC-KD) and a dominant negative construct of dynamin2 (Dyn2-K44A), which

efficiently inhibits the scission of endocytic vesicles by dynamin2. Neither of these treatments inhibited GEM internalization, but both inhibited the endocytosis of fluorescence-labeled transferrin, a marker for clathrin-mediated endocytosis, effectively (Fig. 3E). We concluded that GEM endocytosis occurs independently of clathrin and dynamin in our cells. To ask, whether GEM internalization was affected by the removal of membrane cholesterol, we incubated cells with nystatin (25 µg/ml) and progesterone (10 µg/ml), which effectively reduced SV40 internalization in our system (Fig. 3E) but did not inhibit GEM

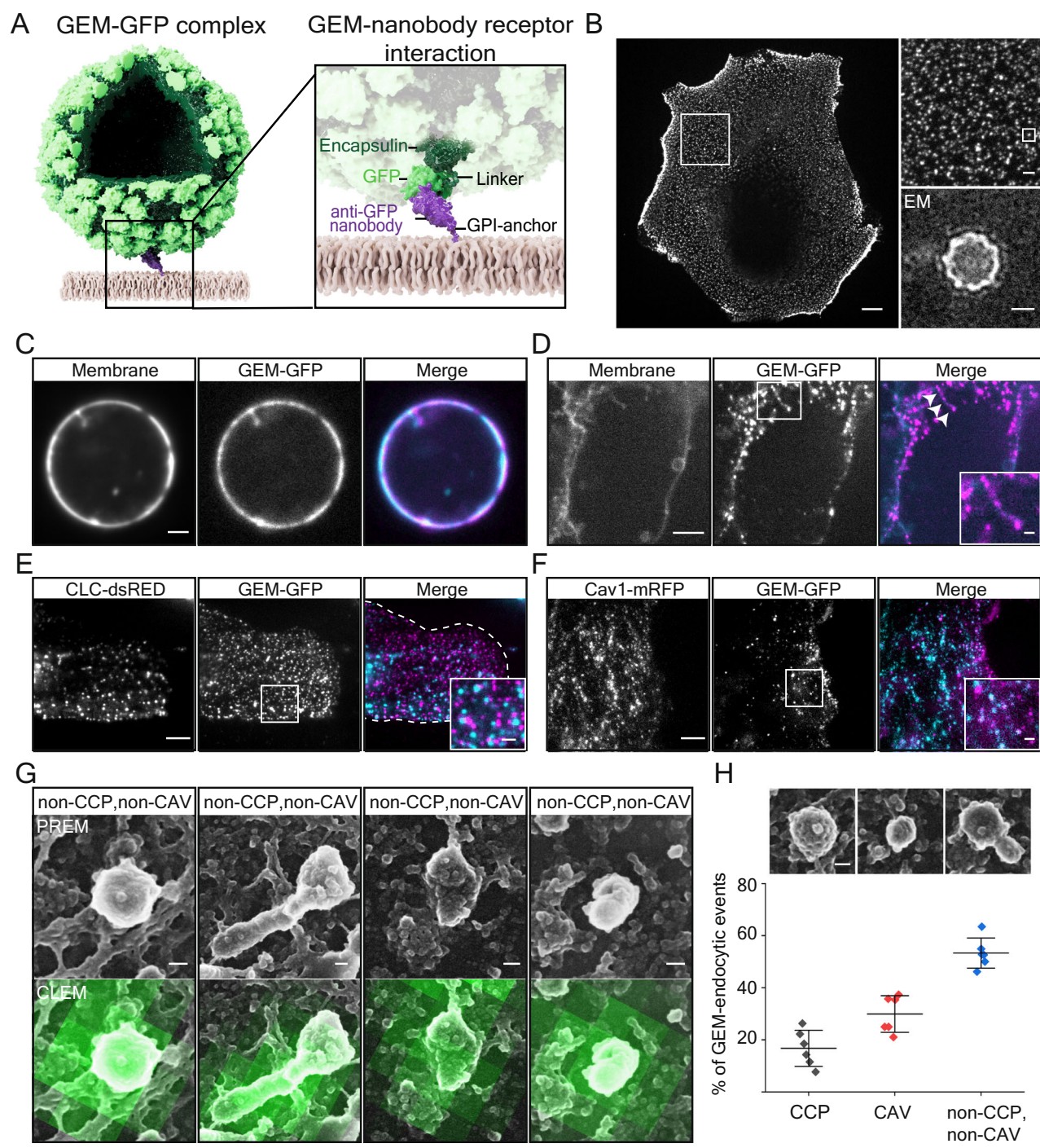

internalization. Similarly, the endosomal acidification inhibitor Bafilo-mycinA, which inhibits SV40 endocytosis[25] (Fig. 3E and Supplementary Fig. S8E, F), did not inhibit GEM internalization. The actin poly-merization inhibitor Cytochalasin D has a slight effect on GEM inter-nalization, but less than for the SV40 virus (Fig. 3E). We concluded that GEMs are not internalized by clathrin-mediated endocytosis or a cholesterol-dependent endocytic pathway, that endosome acidifica-tion is not important in GEM endocytosis. The strongest effect we observed for interference with actin polymerization, suggesting that actin may be involved to some extent.

## Adhesion energy controls endocytosis

Since we found that multivalent binding of our GEM via its surface GFP moieties to lipid-anchored nanobodies was sufficient to allow

adhesion, membrane deformation and endocytosis, we explored whether the affinity of the individual ligand-receptor interaction and thus the adhesion energy of the particle to the membrane would be important in this process. To do so, we created 7 GPI-anchored anti-GFP nanobodies with individual binding affinities increasing from the μM to the pM range (Fig. 4A). Their successful incorporation into the extracellular leaflet of the plasma membrane of cells was verified by the binding of recombinant EGFP to the membrane of transfected cells (Supplementary Fig. S5A, C). Further, our GEM particles bound to cells in an affinity-dependent manner, with binding levels close to back-ground level for the two lowest affinity nanobody constructs (Sup-plementary Fig. S5A, B).

We next investigated whether the strength of the ligand-receptor interaction had an influence on membrane curvature generation.

**Fig. 2 | A polyvalent virus-like-particle lipidic-receptor system for endocytosis.**
**A** Schematic representation of the synthetic system. Shown is a genetically encoded nanoparticle (GEM) assembled from 180 copies of the encapsulin protein (dark green) coupled to GFP (light green) scaffold. A GPI-anchored anti-GFP nanobody (purple) inserted into the membrane (beige) serves as receptor. **B** Fluorescence micrograph of GEMs binding to the cell membrane of CV1 cells. Scale bar is 10 μm. Insets: (upper) magnified region of the GEM-GFP decorated membrane from the overview emphasizing monodisperse binding. A single particle is marked with a box. Scale bar is 2 μm. (lower) Transmission electron micrograph of purified GEM. Scale bar is 15 nm. Experiments have been repeated twice with similar results. **C** Fluorescence micrograph of GEMs bound to Giant Plasma Membrane Vesicles (GPMVs) of CV-1 cells expressing GPI-anchored nanobody. Cells were incubated with 0.45 nM GEMs for 1 h at RT before imaging at the equatorial plane on a spinning disk confocal microscope. Experiments have been repeated twice with similar results. Scale bar is 2 μm. **D** Fluorescence micrograph of GEMs bound to energy-depleted CV1 cells expressing GPI-anchored anti-GFP nanobody that were starved of cellular energy by 30 min incubation in starvation buffer (PBS+/+ supplemented with 10 mM 2-deoxy-D-glucose and 10 mM NaN3) followed by 1 h incubation with 2 μg of purified GEMs in starvation buffer and imaged live on a spinning disk confocal microscope. DiI membrane dye was added 10 min prior to imaging at 1 mg/ml final concentration. Experiments have been repeated twice with

similar results. Scale bars are 5 μm and 1 μm for the inset. **E** Fluorescence micrograph of GEMs bound to CV1-cells expressing Clathrin-light-chain-dsRED incubated for 10 min with 2 μg of GEMs before live imaging on a TIRF microscope. Experiments have been repeated three times with similar results. Scale bars are 5 μm and 1 μm for inset. **F** Fluorescence micrograph of GEMs bound to CV1-cells expressing Caveolin-1-mRFP incubated for 10 min with 2 μg of GEMs before live imaging on a TIRF microscope. Experiments have been repeated three times with similar results. Scale bars are 5 μm and 1 μm for inset. **G** Correlative confocal fluorescence platinum-replica electron microscopy micrographs of plasma membrane sheets generated after unroofing cells incubated with GEMs. Shown are 4 representative intracellular plasma membrane structures colocalizing with GEMs bound to the outside of cells that are neither positive for clathrin (as shown by antibody-staining) nor caveolae (based on distinct caveolae protein coat). Scale bars are 50 nm. Electron microscopy micrographs are on top, same field of view with correlative GFP fluorescence of the GEMs at the bottom. **H** Top: Example platinum replica electron microscopy micrographs of a typical clathrin-coated pit, caveola and clathrin/caveolin double-negative invagination. Bottom: Quantification of colocalization of GEM fluorescence with endocytic structures. Means ± S.D. for 6 cells from $n = 2$ independent experiments. Scale bar is 50 nm. Overview images are provided in Supplementary Fig. S2 and Supplementary Movie 2. Source data are provided as a Source Data file.

When we added GEMs to GPMVs derived from cells expressing our panel of nanobody receptors, we observed membrane tubulation occurring only for the five receptors with the highest binding affinities (Fig. 4B), even though we observed GEMs to bind readily to GPMVs of lower affinity as well. When we repeated these experiments in cells depleted of energy that likewise expressed our panel of nanobody receptors, we found the same pattern (Fig. 4C). To our surprise, unlike for binding, we noticed a yes or no pattern for membrane deformation both in cells and in GPMVs rather than a linear increase with increasing affinity. It seemed that unlike binding, which was observed for all receptor affinities by the GEMs, likely due to the multivalency resulting in high avidity, membrane deformation rather followed a yes or no phenotype with a threshold at 600 nM (Fig. 4B, C). We then performed endocytosis FACS assays in our cells for the entire panel of receptor affinities and found that both the amount of internalized particles in cells as well as the amount of cells internalizing GEMs dropped with decreasing affinity (Fig. 4D, Supplementary Fig. S6). Importantly, we noted a striking decrease for the two samples with the lowest binding-affinity receptors, just like we had observed for membrane deformation, strongly suggesting the existence of a threshold in the adhesion-energy necessary for the polyvalent globular binders to efficiently internalize into cells. It seemed that once a threshold in adhesion-energy was met, particles would deform membranes and become internalized.

To test, whether this striking change in endocytic capacity resulted from a pathway switch, we investigated the colocalization of GEMs in all tested affinities for colocalization with clathrin light chain and caveolin1. We found uniformly low colocalization with both clathrin and caveolin1, while we could observe a slight increase in colocalization with caveolin1 with decreasing affinity (Supplementary Fig. S7). Furthermore, we tested GEM endocytosis for all affinities in comparison to transferrin and SV40 for their susceptibility to inhibitor treatment. Again, the knockdown of CHC and overexpression of dominant negative dynamin2K44A significantly reduced the internalization of fluorescence-labeled transferrin, but not of GEMs binding to any affinity receptor (Fig. 4E). Importantly, we controlled for the amount of surface expressed transferrin receptor and GPI-anchored nanobody receptor upon CHC knockdown (Supplementary Fig. S8). Finally, we tested, whether endocytosis was dependent on cholesterol or endosomal acidification for any receptor affinity in comparison to SV40 and found that neither treatment with nystatin/progesterone for cholesterol reduction nor inhibition of endosomal acidification by BafilomycinA, both of which reduced SV40 endocytosis, did so for GEMs at any receptor affinity (Fig. 4E).

We moved on to generate a physical model of the interaction of the GEMs with plasma membranes to relate receptor affinities to adhesion energies and understand how the resulting change in adhesion energy may control membrane deformation. The curvature generation and membrane tubulation induced by the globular GEMs can be understood from the interplay of membrane bending and particle adhesion energies. The multivalent binding of the GEMs leads to an effective adhesion potential

$$V(l) = \frac{\Delta G(l)}{A} \qquad (1)$$

where $\Delta G(l)$ is the binding free energy of a GFP-nanobody complex, and $A$ is the membrane area per complex. This binding free energy depends on the distance $l$ between the GEM surface and the membrane (Fig. 5A, B), because GFP-nanobody binding is only possible for distances that are compatible with the dimensions and anchoring flexibility of the protein complex. From the dimensions and linker attachment sites of the GFP-nanobody complex, and from the flexibility of these linkers, which allow also for tilting of the complex (see Fig. 5A, B), we estimate the adhesion potential as Gaussian function[31]

$$V(l) = -U \exp\left[-(l-l_0)^2/2\sigma\right] \qquad (2)$$

with preferred binding separation $l_0 = 8$ nm from particle surface to membrane midplane and a standard deviation $\sigma$ of 1 nm (estimated as in Fig. 5A). The depth of this potential is the adhesion energy

$$U = \frac{k_B T}{A} \ln\left[[R]/\xi K_D\right] \qquad (3)$$

where $k_B T$ is the thermal energy, $[R]$ is the surface concentration of the nanobodies in the membrane, $K_D$ is the dissociation constant of the soluble (non-anchored) complex, and $\xi$ is a "conversion length" from 3D binding of the soluble complex to 2D binding of the anchored complex at preferred separation $l_0$[31]. Because the radius of the GFP-free GEM scaffold particle is 15 nm[27], the radius of a membrane vesicle wrapping a single particle is $r = 15 + 8$ nm $= 23$ nm, and the vesicle area per protein complex is $A = 4\pi r^2/180 = 37$ nm$^2$.

Membrane deformation through particle wrapping becomes energetically favorable when the adhesion energy of wrapping increases to a level above the cost of membrane bending[32,33], i.e. above the threshold level $U_t = 2\kappa/r^2$. Below this threshold, membrane bending

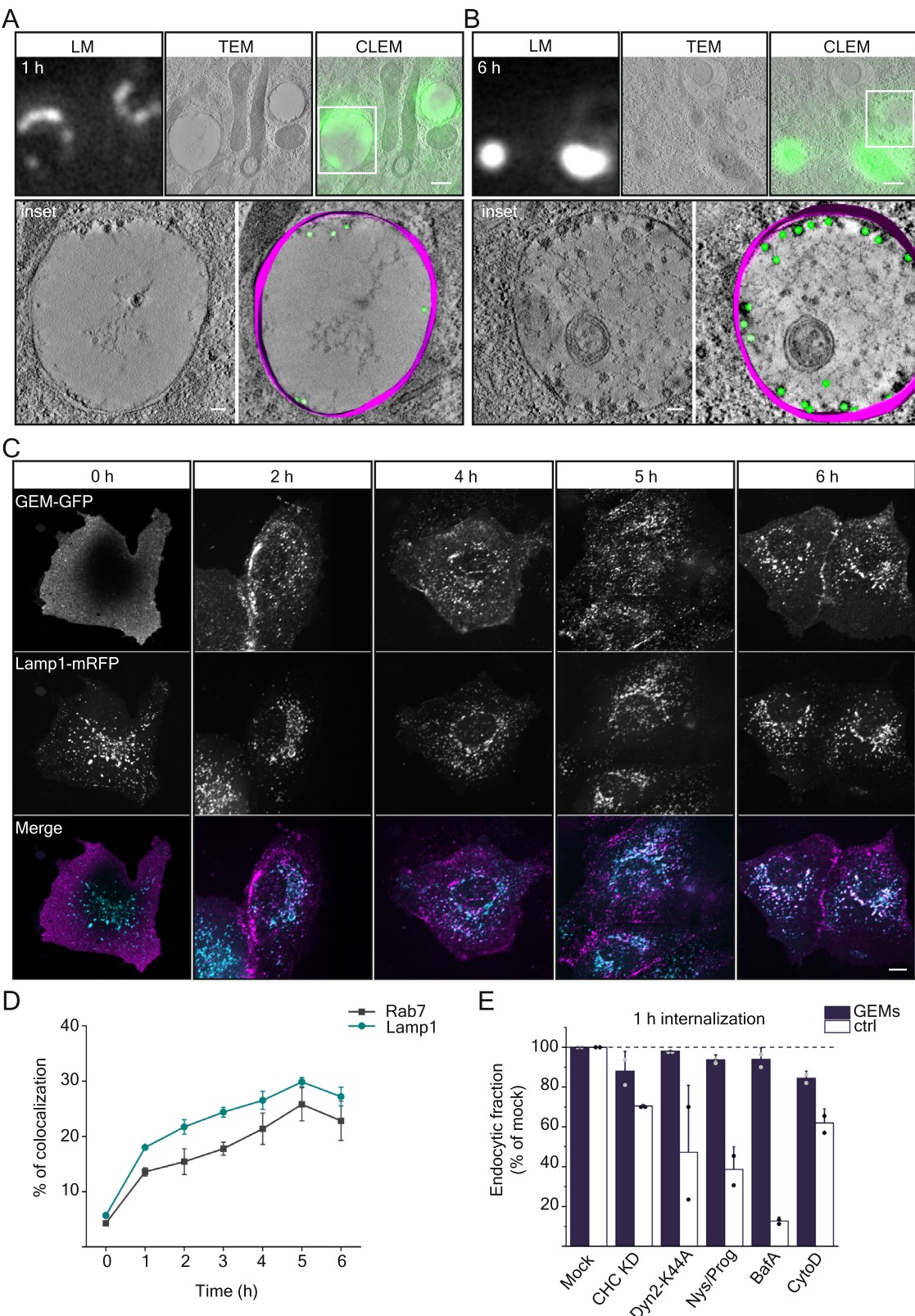

and thus particle wrapping is energetically unfavorable and does not occur, above it, it does. The wrapping threshold in our cell experiments occurs around $K_D = 1000$ nM (between 600 nM and 3800 nM, see Fig. 5B), which corresponds to a threshold level $U_t$ with $[R]/\xi = 16{,}300$ nM according to Eq. (3) for the typical membrane bending rigidity $\kappa = 20$ $k_B T$. For adhesion energies $U > U_t$, both wrapping of single particles (Fig. 5D, left) and the joint wrapping of particles

by membrane tubules (Fig. 5D, right) can occur. We found that the sum of bending and adhesion energies is clearly lower for joint particle wrapping, which explains the observation of membrane tubulation in energy-depleted cells and GPMVs. The calculated energy gain for the joint wrapping in tubules, compared to individual wrapping, is about 10 to 40 $k_B T$ per particle for the typical membrane bending rigidity $\kappa = 20$ $k_B T$[34] (Fig. 5C). We assume that in energy-depleted cells and

**Fig. 3 | GEMs are endocytosed and are trafficked through the endolysosomal pathway. A, B** Correlative fluorescence light microscopy and transmission electron microscopy of GEMs internalized in CV-1 cells. **A** Timepoint 1 h after binding. **B** Timepoint 6 h after binding. Each panel top from left to right: Fluorescence micrograph of GEMs; transmission electron micrograph of same region; correlative images. Each panel bottom left: Transmission electron micrograph of inset above. Each panel bottom right: Volumetric 3D-reconstruction of electron micrographs. GEMs emphasized in green, membrane emphasized in purple. Experiments have been repeated twice with similar results. Scale bars are 500 nm for overview and 100 nm for insets. **C** Fluorescence micrographs from a time-course experiments of endocytosis showing the distribution of GEMs in CV1 cells expressing anti-GFP nanobody and Lamp1-mRFP. Cells were incubated with 2 μg of GEMs for the

indicated time points at 37 °C before live imaging on a spinning disk confocal microscope. Scale bars are 10 μm. **D** Quantification of colocalization between GEMs and Lamp1-mRFP and between GEMs and Rab7-mRFP from timepoints indicated in **C** Means ± s.e.m., n = 3 independent experiments. **E** Quantification of GEM endocytosis upon treatment with genetic (siRNA against clathrin-heavy-chain and expression of dominant negative Dyn2-K44A) or chemical inhibitors (Nystatin/Progesterone, BafilomycinA and Cytochalasin **D**) as compared to mock treatment of controls (Transferrin endocytosis for siRNA against CHC and overexpression of DynK44A; SV40 endocytosis for Nystatin/Progesterone, BafilomycinA and Cyto-chalasinD). Mean fluorescence intensity ± S.D. was determined from flow cytometry measurements of 6811–27,733 cells from $n = 2$ independent experiments. Source data are provided as a Source Data file.

GPMVs, the machinery recognizing individually wrapped particles in small early endocytic invaginations (Fig. 2G) is not active. In the absence of cellular energy for endocytosis to progress after partial wrapping, particles accumulated in elongated tubules because it is energetically favorable. In native cells in contrast, we assume that as soon as particles deform the membrane to wrap around themselves they are recognized by an unknown cellular machinery and internalized. Tubulation thus occurs in GPMVs and energy-depleted cells above the threshold leading to membrane deformation, in intact cells, particles are quickly internalized as soon as they become wrapped.

## Discussion

We here developed a system to test the interplay between receptor affinity and multivalent binding of globular particles such as viruses in the induction of membrane deformation and endocytosis. Our system, based on virus-like, icosahedral, genetically encoded multimeric particles (GEMs) and a portfolio of lipidic receptors covering seven orders of magnitude of affinity, allowed us to directly ask: i) how adhesion energy influences the capability of particles to deform cellular membranes and ii) if this is sufficient to induce internalization into cells. We find that indeed a threshold adhesion energy exists for both membrane deformation and endocytosis and strikingly, that this seems to be the same. We find that a $K_D$ of ~1 μM allows for our GEMs to become internalized. The particles follow a clathrin- and dynamin-independent pathway into the cell after forming early endocytic structures devoid of both clathrin and caveolin1. They then become transported through early endosomes to lysosomes in a bafilomycin-independent manner.

It has long been known that several multivalent lipid binders including viruses, bacterial toxins and lectins are capable of mediating their internalization in a clathrin-independent manner. The unifying principle behind clathrin-mediated endocytosis is clear. On the other hand, clathrin-independent processes remain difficult to define[3,4,35–37]. We here reconstituted a synthetic cell biological model for endocytosis built on GEM particles studded with 180 GFPs and anti-GFP nanobody receptors that are anchored in the plasma membrane via a GPI-anchor. Using this reductionist approach, we could isolate the role of the affinity of the receptor and thus the adhesion energy of the particle in membrane deformation and endocytosis. We find a threshold adhesion energy for our GEMs that, when exceeded, leads to their internalization. It is thus safe to assume that our observations are the result of an isolated biophysical mechanism. There have been reports of other cellular processes that become active upon overcoming a mechanical force threshold, i.e. the molecular clutch between the cytoskeleton and extracellular matrix that depends upon talin stretching as a mechanosensor[38–40].

Clearly, biophysical processes at membranes play important roles in cell biology, especially when it comes to membrane shaping and deformation[5,6,35,36,41–45]. Furthermore, line tension and protein crowding can result in membrane scission[46,47]. When we let GEMs bind to GPMVs or membranes of energy-depleted cells, we found that they deformed membranes into tubular structures continuous with the bounding membrane, but did not observe any vesicles inside GPMVs

nor cells. In intact cells, we in contrast did not see membrane tubules. GEMs rather formed small early-endocytic structures that seemed to vesiculate quickly after membrane deformation. This supports a scission mechanism that depends on a cellular machinery rather than a mechanism based purely on biophysical principles. We thus speculate that once the membrane wraps around the ~40 nm diameter particle, this membrane invagination is recognized by the cell, recruiting factors required for vesiculation, scission and transport to endosomes and lysosomes. One such factor may be actin, as the assembly of actin-polymerizing molecules on artificial membranes can lead to abscission and vesicle formation in in vitro assembled systems[3,48] and actin is known to be a central player in clathrin-independent endocytosis[49,50]. BAR-domain proteins like endophilin likewise can mediate scission of clathrin-independent carriers[51–54]. In the future it will thus be of great interest to identify the cellular factors recruited to the highly curved early endocytic carriers that mediate vesicle scission and intracellular transport to endosomes. These factors may be central to many uptake processes of lipid-binding particles.

Several molecular mechanisms to generate curvature on membranes and later endocytosis are known. Lateral aggregation of proteins at membranes can lead to membrane deformation[55] and the formation of protein condensates at membranes can do so as well[56–59]. The multivalent binding of SV40[60] can deform membranes[15], likewise bacterial toxins such as CTxB[14–16,18] and other, small or globular lipid binders[17,47,61]. However, no tractable system exists that allows for a systematic investigation of the role of particle size, receptor affinity or valency in membrane deformation. Our system of GEMs and lipid-linked nanobody receptors offer such experimental control and could be extended further. On the one hand, GEMs exist in a large variety of sizes and geometries[62], presenting the opportunity to investigate the requirements to force membranes into specific curvatures and to refine biophysical models[47]. On the other hand, the synthetic nature of the GEMs together with our physical model allow for precise tuning of individual parameters. The binding sites of the GEM, being GFP proteins attached through a linker, are flexible. The flexibility requires high individual receptor affinities for membrane deformation. In contrast, the SV40 particle with a stiff arrangement of binding sites requires merely low μM affinities in their lipidic receptors for membrane deformation and internalization[15,23,60,63].

In general, the globular SV40 and the likewise multivalent bacterial toxin CTxB belong to the best studied multivalent lipid ligands for membrane deformation. Strikingly, the organization of binding sites in SV40 and CTxB are arranged in a very similar manner, and rearrangement of binding sites can lead to a loss of membrane deformation capacity for CTxB[61] and other lectins[61]. This suggests a strong mechanistic role of binding site geometry in membrane deformation and endocytosis, supporting the notion that nanoscopic application of adhesion energy is required for membrane deformation. Our system of GFP-studded ligands and lipid-anchored nanobody-receptors, besides creating globular binders of differing valency and size, could be extended to pair small multivalent particles of controlled valency with receptors of controlled affinity. Such a system would be

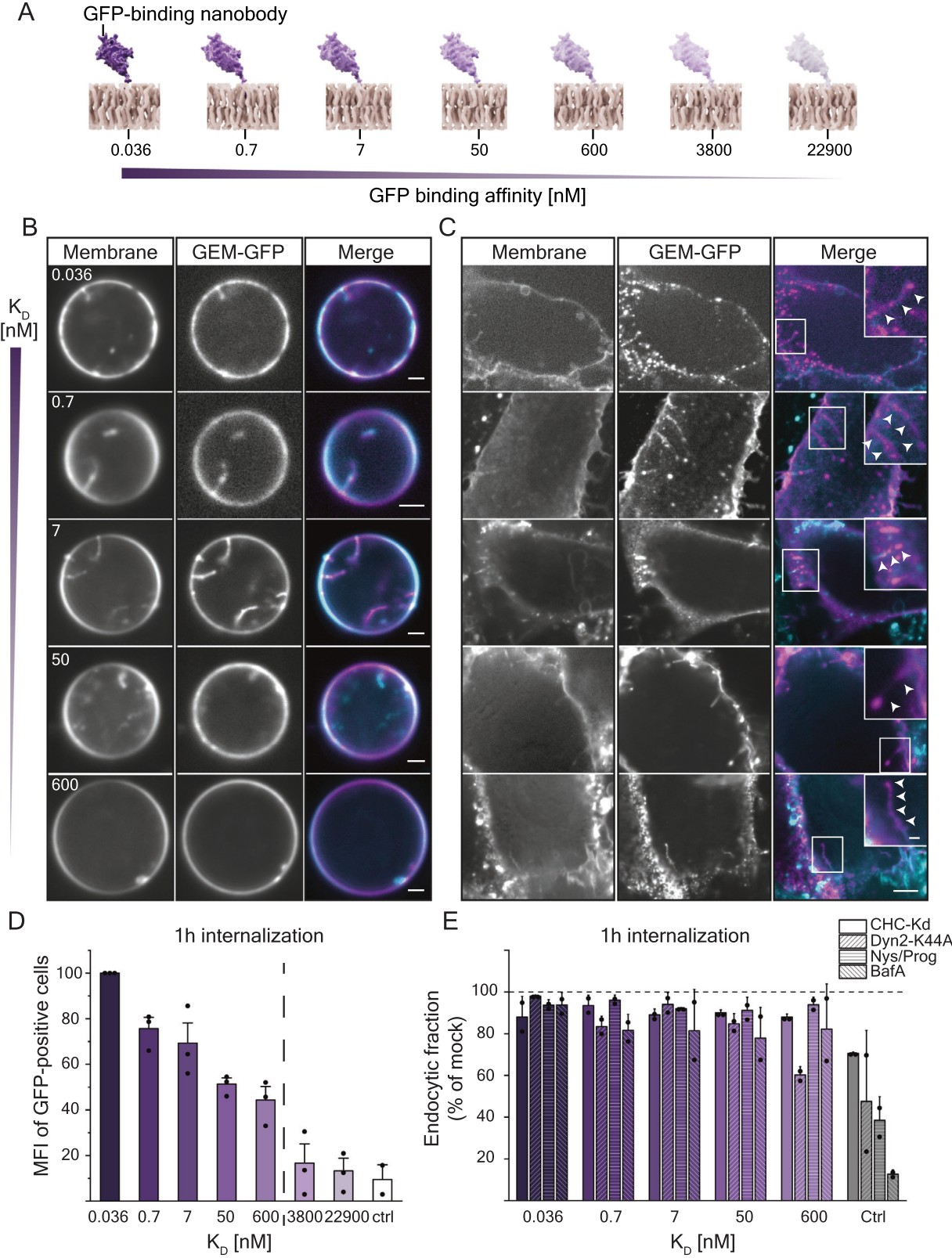

especially powerful in in vitro reconstituted systems, where it would allow for a comprehensive investigation of how binding-affinity, valency and geometry act together in membrane deformation, lipid-mediated endocytosis and membrane domain formation in general[64–66]. This will allow the field to overcome a bottleneck in tractable experimental systems to investigate fundamental biophysical processes at membranes that control cellular function.

# Methods

## Materials

Lipids were purchased either from Avanti Polar Lipids: DOPC (1,2-dioleoyl-sn-glycero-3-phosphocholine), from Enzo Life Sciences: GM1 (Ganglioside GM1. sodium salt (bovine brain)), GD1a (Ganglioside GD1a. disodium salt (bovine brain)), GD1b (Ganglioside GD1b. disodium salt (bovine brain)), GT1b (Ganglioside GT1b. trisodium salt

**Fig. 4 | Adhesion energy controls membrane deformation and endocytosis of GEMs.** **A** Schematic representation of GPI-anchored nanobody constructs with decreasing binding affinity expressed in the outer membrane of cells used in this study. **B** Fluorescence micrographs of GEMs bound to GPMVs harvested from CV1 cells expressing the panel of GPI-anchored anti-GFP nanobody constructs as indicated and subsequently incubated with 0.45 nM GEM-GFP particles for 1 h at RT before imaging at the equatorial plane on a spinning disk confocal microscope. Experiments have been repeated twice with similar results. Scale bars are 2 μm. **C** Fluorescence micrographs of GEMs bound to energy-depleted CV-1 cells expressing the panel of GPI-anchored anti-GFP nanobodies. CV1 cells were starved of cellular energy by 30 min incubation at 37 °C in starvation buffer (PBS+/+ supplemented with 10 2-deoxy-D-glucose and 10 mM NaN3) followed by 1 h incubation with 2 nM of GEM-GFP particles in starvation buffer at 37 °C and imaged live on a spinning disk confocal microscope. DiI membrane dye was added 10 min prior to imaging at 1 mg/ml final concentration. Experiments have been repeated twice with similar results. Scale bars are 5 μm and 1 μm for insets. Arrows mark VLP-filled

membrane invaginations. **D** Quantification of GEM-GFP endocytosis as a function of receptor affinity as determined by flow cytometry measurements of the mean cell-associated fluorescence after acid wash. Mean fluorescence intensity ± s.e.m. was determined from flow cytometry measurements of 846–9538 cells/sample from n = 3 independent experiments. **E** Quantification of GEM-GFP endocytosis as a function of receptor affinity and upon treatment with genetic (siRNA against clathrin-heavy-chain and expression of dominant negative Dyn2-K44A) or chemical inhibitors (Nystatin/Progesterone and BafilomycinA) as compared to mock treatment of controls (Transferrin endocytosis for siRNA against CHC and over-expression of DynK44A; SV40 endocytosis for Nystatin/Progesterone and BafilomycinA). Endocytosis was determined by flow cytometry measurements of the mean cell-associated fluorescence after acid wash. Mean fluorescence intensity ± S.D. was determined from flow cytometry measurements of 1026–29,803 cells from n = 2 independent experiments. Source data are provided as a Source Data file.

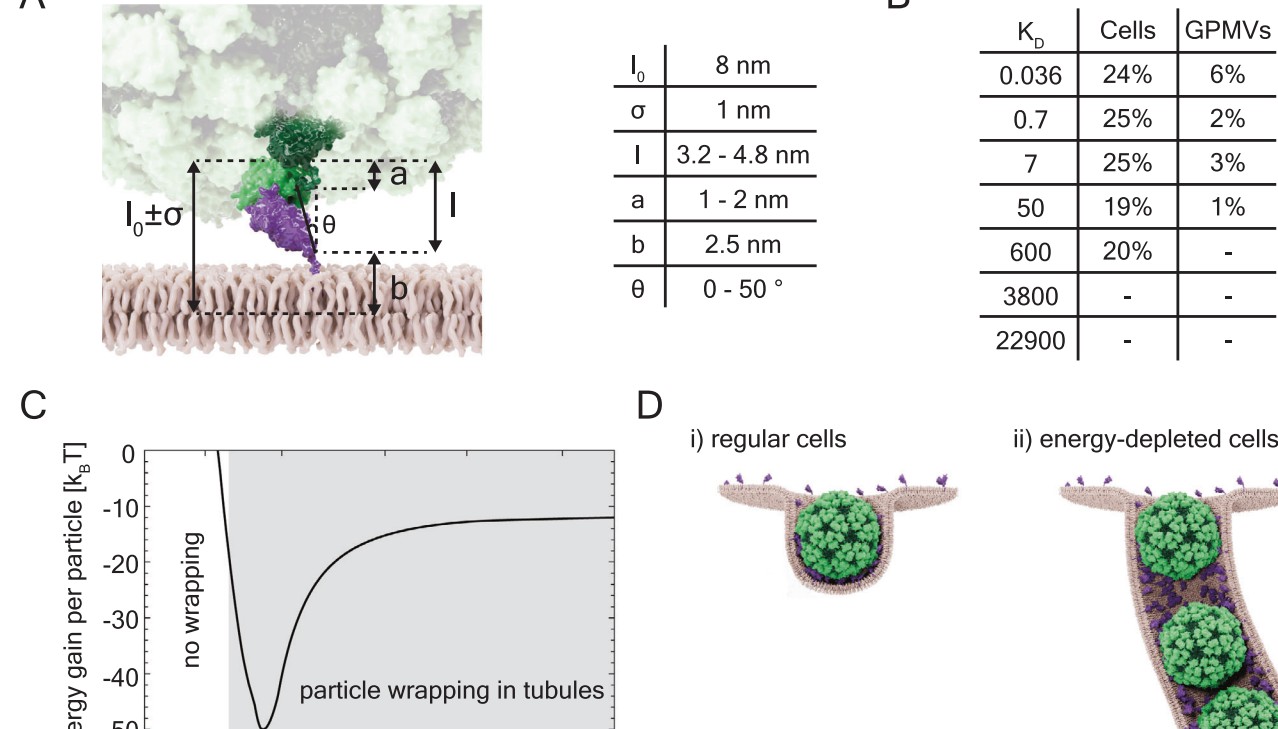

**Fig. 5 | Theoretical modeling of GEM-GFP wrapping and tubulation.** **A** Schematic representation of the geometric parameters considered in the modeling for the length and tilt angle variations (cartoon) and the corresponding estimative numerical values (table): b = estimated distance from membrane midplane to nanobody C-terminus to which the GPI-anchor is attached; a = estimated vertical extensions of unstructured 12-residue peptide linker connecting the GFP N-terminus to the GEM surface; Θ = estimated tilt angle of the complex, i.e of the axis (with length 4.8 nm) connecting the linker attachment sites at the nanobody C-terminus and GFP N-terminus, relative to the membrane normal; l = projected vertical extensions 4.8 nm Cos[Θ] of the complex corresponding to tilt angle estimates. These length estimates and variations lead to the mean distance $l_0 = 2.5 + 1.5 + 4$ nm = 8 nm and standard deviation σ = −1 nm. **B** Table of the

percentage of energy-depleted cells and GPMVs containing GEM-filled tubular invaginations from the total amount of cells/GPMVs for the corresponding binding affinities. **C** Energy gain per central GEM particle in a tubule, compared to individual wrapping of the particle, as a function of the binding affinity. Above the wrapping threshold around Kd = 1000 nM, particle wrapping in tubules is energetically clearly favorable. **D** Schematic representation of the two membrane wrapping cases: i) GEM wrapping in live cells leading to cellular endocytosis and ii) GEM wrapping resulting in the formation of long tubules in energy-depleted cells, illustrated here for an intermediate conformation with three particles close to the wrapping threshold where the particles are only partially wrapped. Modeled minimum-energy conformations for GEM wrapping are shown in Supplementary Fig. S9. Source data are provided as a Source Data file.

(bovine brain)) and from ThermoFisher Scientific: β-BODIPY™ FL C12-HPC (Invitrogen), DiIC18(3) stain (Invitrogen). Transferrin from Human Serum, Alexa Fluor™ 488 Conjugate (Invitrogen) was purchased from ThermoFisher Scientific, Purified recombinant Enhanced Green Fluorescent Protein (EGFP) was purchased from Chromotek.

Bafilomycin A1 was purchased from InvivoGen, Nystatin and Progesteron were purchased from Sigma-Aldrich. Clathrin Heavy Chain siRNA (Human CLTC, sequence: GGUUGCUCUUGUUACG, ID: s475) and Negative Control#1 siRNA Silencer Select were purchased from ThermoFisher Scientific.

## Virus-like-particles

Polyomavirus-like-particles were assembled from purified VP1 proteins obtained from Abcam, namely Simian Virus 40 (ab74565), mouse polyomavirus strain RA (ab74571) and JC polyomavirus (ab74569), according to the manufacturer's specifications. Alexa Fluor 647-NHS was covalently coupled to the assembled virus-like-particles in 0.2 M NaHCO3 at pH 8.3 using a 10-fold molar excess of the dye relative to VP1 protein. Unbound dye was removed by two subsequent washing steps on pre-equilibrated Zeba columns (40 KDa cut-off, Thermo Scientific) in PBS buffer.

## Gene cloning and plasmids

Nanobody sequences (Clone IDs: LaG16-G$_4$S-2, LaG-16, LaG-21, LaG-17, LaG-42, LaG-18 and LaG-11 from[28]) were codon-optimized for expression in mammalian cells and cloned into a Twist Amp High-Copy vector after gene synthesis (Twist Bioscience), incorporating BamHI and XhoI restriction sites at 5′ and 3′ ends, respectively. Next, the nanobody sequences were subcloned into a pEGFP-N1 GPI-GFP vector in between the LPL-signal peptide and the GPI-anchor, replacing the GFP sequence. Successful insertion was verified by gene sequencing (Microsynth AG). GEM-GFP sequence[26] was codon-optimized for expression in *Escherichia coli* and cloned into a pET-29b(+) vector in between the NdeI and XhoI restriction sites after gene synthesis (Twist Bioscience). Lamp1-mRFP, Lamp1-EGFP, Rab7-mRFP, Clathrin light chain–mRFP, Caveolin1-mRFP and cytosolic RFP were a kind gift from the Ari Helenius laboratory. RFP Dynamin2 K44A and RFP Dynamin-2 Wild Type were a gift from the Jennifer Lippincott-Schwartz laboratory (Addgene plasmid # 128153 and Addgene plasmid # 128152)[67].

## Recombinant expression and protein purification

GEM-GFP particles were expressed under a T7 promoter in *E.coli* BL21 strain in complex autoinduction medium (1% N-Z-amine AS, 0.5% yeast extract, 25 mM Na$_2$HPO4, 25 mM KH$_2$PO$_4$,50 mM NaH$_4$Cl, 5 mM Na$_2$SO$_4$, 2 mM MgSO$_4$, 0.2x trace metals, 0.5% glycerol, 0.05% glucose, 0.2% α-lactose, 30 μg/ml kanamycin) at 37 °C for 4 h, followed by further incubation at 21 °C for 72 h. The bacterial pellet was resuspended in lysis buffer (50 mM Phosphate buffer, 50 mM NH$_4$Cl, 40 mM imidazole, 700 mM NaCl, 10% Glycerol, 1 mg/ml lysozyme, 10 μg/ml DNAse I, protease and phosphatase inhibitor cocktail (Thermo Scientific) at pH 7) at 4 °C for 30 min and subsequently heated up at 55 °C for 30 min. The lysate was sonicated and cleared by centrifugation (7000 × *g*, 40 min, 4 °C). The supernatant was added to a pre-equilibrated Ni-NTA-bead gravity flow column, washed with washing buffer (50 mM Phosphate buffer, 50 mM NH$_4$Cl, 40 mM imidazole, 700 mM NaCl, 10% Glycerol, protease and phosphatase inhibitor cocktail at pH 7) and eluted with elution buffer (50 mM Phosphate buffer, 50 mM NH$_4$Cl, 500 mM imidazole, 700 mM NaCl, 10% Glycerol, protease and phosphatase inhibitor cocktail at pH 7). The elution was then dialyzed overnight at 4 °C into protein buffer (50 mM Phosphate buffer, 50 mM ammonium chloride, 700 mM NaCl, 5% Glycerol, protease and phosphatase inhibitor cocktail at pH 7). Next, a size-exclusion chromatography run on a Superdex 200 Increase 10/300 GL column was performed in protein buffer. The fractions eluted in the void volume of the column were verified to contain GEM-GFP proteins by SDS-Page gel and by MALDI mass spectrometry. The fractions were then pooled, concentrated on a Amicon Ultra 100 K (Merk Millipore) concentrator, stored at 4 °C and used in the first two weeks after purification.

## Cell culture and transfections

CV1 (ATCC CCL-70) and NRK49F (ATCC CRL-1570) cells were cultured in DMEM (Gibco) supplemented with 10% fetal bovine serum (Corning), 1 mM GlutaMax (Gibco). Cells were regularly tested for mycoplasma contamination.

Cells were transfected by electroporation using a Neon transfection system kit (Thermo Fischer) according to the manufacturer's specifications. In brief, cells were detached with Trypsin (Gibco) and washed one time in PBS before resuspension in R-buffer. Cells were then mixed with either 1 μg (single transfection) or 0.5 μg (double transfection) of each plasmid used and transfected in a 10 μl Neon pipette tip with two electric pulses at 1050 V for 30 ms. After transfection, cells were plated onto 12-well plates (for flow cytometry) or on 18-mm glass coverslips, thickness 1.5 (VWR, Cat. – No. 631-0153) (for microscopy) and grown for 24 h in medium at 37 °C before use.

For knock-down experiments, cells were transfected with Poly-Fect Transfection Reagent (Qiagen) according to the manufacturer's specifications. In brief, $10^5$ cells were plated in 6-well plates one day prior to transfection. On transfection day, 4 μg of siRNA was diluted in OptiMEM and subsequently mixed with PolyFect Transfection Reagent. After 20 min incubation at room temperature, the mixture was added to the cells and further incubated for 48 h before measurement.

## Binding assays

Cells were plated a day prior to experiments on 18-mm cover glass, thickness 1.5 (VWR, Cat. – No. 631-0153). For the EGFP/GEM-GFP binding assay, cells were transfected with the nanobody constructs a day prior to the binding assay, as described in the previous section. On the measurement day, cells were incubated at 4 °C for 20 min to stop endocytosis and further incubated with 2 μg/ml of either VLPs, recombinant EGFP or purified GEM-GFPs at 4 °C for 30 min. Then, the cells were fixed with 4% PFA, 0.2% GA in PBS at RT for 20 min. The cells were washed with PBS and the fixation solution was quenched in 50 mM NH$_4$Cl in PBS at RT for 30 min and imaged on a spinning disk confocal microscope.

## Endocytosis assay and inhibitor treatments

We quantified the GEM-GFP endocytosis amounts for all the different binding affinity nanobody-GPI constructs by performing quantitative endocytosis assays using flow cytometry measurements. In brief, cells were co-transfected with the nanobody constructs and a cytosolic mRFP-marker to select the positively transfected cells a day prior to the endocytosis assay, as described in the previous section. On the measurement day, the cells were washed with PBS and resuspended in fresh medium. For inhibitor endocytosis assays, cells were resuspended in either fresh medium supplemented with DMSO (control) or fresh medium supplemented with inhibitors as follows: 100 nM BafilomycinA/ 25 μg/ml nystatin and 10 μg/ml progesterone/ 10 μM Cytochalasin D followed by 1 h/ overnight/ 10 minutes incubation at 37 °C, respectively. Either 10 μg/ml Transferrin AF-488, 2 μg/ml GEM-GFP or 2 μg/ml VLPs were added to the cells in medium (endocytosis assay) or in medium supplemented with inhibitors (inhibitor assay) and further incubated for 1 h at 37 °C. Next, cells were washed 3× in acid buffer (0.5 M glycine in PBS, pH 2.2) to remove all surface-bound fraction of VLPs/GEMs and 1× in PBS before detaching with trypsin or fixation (for the BafilomycinA microscopy control). Cells were resuspended in fresh medium and measured with a BD FACSCanto Flow Cytometry System. The gating strategy employed in the flow cytometry data analysis is explained in Supplementary Fig. 5. The fixed cells prepared in the Bafilomycin A control experiments were quenched with 50 mM NH4Cl and imaged on a spinning disk confocal microscope.

For Transferrin Receptor and Nanobody-GPI surface level quantification, cells were transfected as described in the previous section a day prior to the experiment. On the measurement day, the cells were washed with PBS and resuspended in serum-free medium and were incubated at 4 °C for 15 min before addition of either

 

10 µg/ml Transferrin AF-488 or 2 µg/ml purified EGFP and further incubation at 4 °C for 45 min. Cells were then washed ×2 with cold PBS and were detached with accutase. Cells were resuspended in serum-free medium and measured with a BD FACSCanto Flow Cytometry System.

For GEM-GFP binding inhibition experiments, cells were transfected with the 0.036 nM binding affinity nanobody one day prior to experiment as described in the previous section. On the measurement day, 2 µg/ml GEM-GFP were pre-incubated with either 0 (ctrl), 4, 8 or 20 µg/ml of recombinant LaG16 nanobody for 5 min at RT. Then, the mix was added to the cells and they were imaged live on a spinning disk microscope.

## Clathrin- or caveolin-colocalization assays

CV1 cells were co-transfected with the specified GPI-nanobody construct and with either Clathrin-Light-Chain-mRFP or Caveolin1-mRFP as described in the previous section. For polyoma VLP colocalization, cells were transfected with either Clathrin-Light-Chain-mRFP or Caveolin1-mRFP as described in the previous section. Next day, cells were washed 2x with PBS and fresh medium supplemented with 10 mM HEPES was added. Next, 2 µg/ml of either GEM-GFP or polyoma VLPs were added to the cells and incubated for 10 min at 37 °C before imaging live on a TIRF (total internal reflection fluorescence) microscope.

## Pulse-chase assay for intracellular traffic

CV1 cells were transfected with either endosomal or lysosomal markers one day prior to the pulse chase experiments as described in the previous section. The next day, cells were washed with PBS, then resuspended in fresh medium and incubated at 4 °C for 20 min to stop endocytosis. Next, 2 µg/ml of either VLPs or GEM-GFP were added to the cells and incubated further for 30 min at 4 °C to allow for protein binding while endocytosis is inhibited. Then, cells were washed with PBS and resuspended in warm medium supplemented with 10 mM HEPES. Cells were imaged live and right away corresponding to time point t = 0 min. Afterwards, cells were placed at 37 °C and incubated for the respective amounts of time before imaging live on a spinning disk confocal microscope.

## Cellular energy starvation assay

Cellular energy was depleted by incubating CV1 cells in PBS++ supplemented with 10 mM 2-deoxy-D-glucose and 10 mM NaN₃ for 30 min at 37 °C until residual ATP levels dropped to 2.1% according to previous findings[68]. Next, cells were incubated with at least 30 µg/ml of the specified VLPs or 2 nM of GEM-GFP in energy-depletion medium for 1 h at 37 °C. In the last 10 min of incubation, 1 mg/ml of DiI C12 membrane dye was added to the cells for the remaining time. Cells were then imaged live in energy-depletion medium supplemented with 10 mM HEPES on a spinning disk confocal microscope.

## Model membrane systems: Giant Unilamellar Vesicles

GUVs were grown using the electroformation technique as previously described[69]. Lipid mixtures were prepared in a methanol:chloroform solvent to 1 mg/ml final concentration. Next, 5 ul of the mix were spread on each platinum wire of an in-house-built Pt electrode electroformation chamber. An electric current was applied and vesicles were grown in a 300 mM sucrose solution for 1 h at 10 Hz and 2 V at room temperature. The alternating current was then decreased to 2 Hz and 2 V for another 30 min. Once the electroformation procedure was completed, the GUV suspension was dropped onto coverslips that have been pre-incubated with 1 mg/ml BSA solution and washed in PBS. GUVs were subsequently incubated with 10 µg/ml of the specified VLPs for 1 h at room temperature in VLP buffer (10 mM HEPES at pH 6.8, 150 mM NaCl and 2 mM CaCl₂) and then imaged on a spinning disk confocal microscope.

## Model membrane systems: Giant Plasma Membrane-derived Vesicles

GPMVs were isolated from CV1 cells a by chemical vesiculant technique as previously described[70]. Briefly, CV1 cells close to confluency were washed with PBS and incubated with 4 µg/ml of DiI C12 membrane dye in PBS for 10 min at 37 °C. Cells were washed in PBS and resuspended in GPMV buffer (10 mM HEPES, 150 mM NaCl, 2 mM CaCl₂, pH 7.4) supplemented with 10 µM of the vesiculation agent calmidazolium. After 2 h incubation at 37 °C, the supernatant was transferred to an Eppendorf tube and GPMVs were allowed to settle down for 30 min at RT. Finally, 200 µl of the GPMV solution was dropped onto a 8-well imaging chamber containing 200 µl of GPMV buffer. GPMVs were further incubated with 0.45 nM of GEM-GFP protein solution for 1 h at RT and imaged on a spinning disk confocal microscope.

## Western Blot

CV1 cells were transfected with the indicated GPI-anchored nanobody constructs and with Clathrin Heavy Chain siRNA as described in the previous section. After 48 h, cells were detached with trypsin and resuspended in lysis buffer (0.1% Triton-X in PBS) and incubated at 4 °C for 20 min. Cell suspensions were spun down at 20,000 × g for 40 min at 4 °C. The supernatant was collected, denatured and run on a SDS-PAGE 4–12% Bis-Tris gel (Eurogentec, ID-PA4121-010) in MOPS buffer. Blotting was performed by Trans-Blot Turbo (Bio-Rad) with 0.2 µm PVDF membranes (Bio-Rad, IB301002) accordingly to the manufacturer's protocol. Afterwards, the membrane was blocked in TBS supplemented with 5% BSA for 1 h at RT. Next, the membrane was incubated overnight at 4 °C with 1:1000 dilution of anti-CHC antibody (Cell Signaling Technology, P1663) in TBS-T. The next day, the membrane was washed three times in TBS-T and further incubated for 1 h at RT with 1:1000 dilution of secondary goat anti-rabbit HRP antibody (Invitrogen, 31462) in TBS-T. Lastly, the membrane was washed three times with TBS-T and imaged in ECL solution. Next, the membrane was stripped of antibodies in a mild stripping solution for 1 h at RT (200 mM Glycine, 1% SDS, 10% Tween-20 in dH₂O, pH 2.2) before the staining and imaging procedures were performed again with loading control anti-GAPDH antibodies at 1:1000 dilution (Abcam, ab8245).

## Correlative light and electron microscopy

Transfected cells expressing the Nanobody-GPI construct were grown on carbon-coated sapphire discs (3 mm diameter, 50 µm thickness, Wohlwend GmbH, art. 405). Next day, cells were treated with 5 µg/ml GEM-GFP at 4 °C for 20 min and then transferred to 37 °C for the indicated times. After treatment, the samples were high pressure frozen (HPM010, AbraFluid) in their growth medium and freeze substituted (EM-AFS2, Leica Microsystems) with 0.1% uranyl acetate in dry acetone at −90 °C for 40 h. The temperature was then raised to −45 °C with a rate of 4.5 °C/h and the sample were further incubated for 5 h. After rinsing in acetone, the samples were infiltrated with increasing concentrations of Lowicryl HM20 resin (25%, 50%, 75%, 4 h /step and 3 × 10 h in 100%), while raising the temperature to −25 °C. Finally, the samples were UV-polymerized at −25 °C. The sapphire disc was then removed from the resin and 300 nm sections parallel to the block surface were cut and collected on carbon coated mesh grids (S160, Plano). Fluorescence imaging of the sections on the grids was carried out with a widefield fluorescence microscope (Olympus IX81) equipped with a 100 × 1.40 NA Plan-Apochromat oil immersion objective. After post-staining with 2% uranyl acetate in 70% methanol and Reynold's lead citrate, tilt series of the areas of interest were acquired with TECNAI F30 transmission electron microscope (FEI) at 300 kV acceleration voltage using the software package SerialEM[71]. Tomograms were reconstructed using IMOD[72]. Correlation between fluorescence and electron microscopy images was performed with the plugin ec-CLEM[73] of the software platform Icy[74], using features of the sample that could be identified in both imaging modalities.

## Platinum Replica Electron Microscopy (PREM)

NRK49F cells were transfected with the 0.036 nM binding affinity GPI-anchored nanobody plasmid using Lipfectamine 3000 (Thermo Fisher Scientific) according to the manufacturer's instructions. 24 h after transfection, the cells were detached with 1 mM EDTA in PBS, pelleted at $200 \times g$ for 4 min, and resuspended in cellular medium containing 0.3 μg/ml GEMs. Cell suspension was incubated with GEMs for 5 min at 37 °C and inverted every 1 min. 25 mm round coverslips (thickness no. 1.5) were coated with 0.01% (wt/vol) poly-L-Lysine solution (Sigma) for 20 min and cell-GEM suspension was then plated on the coverslips. Cells were attached to the coverslips by centrifugation at $100 \times g$ for 1 min. After attachment cells were incubated at 37 °C for 10 min prior to unroofing and fixation.

Cells were unroofed to obtain plasma membrane sheets as described previously[29,30]. Briefly, cells on coverslips were placed in stabilization buffer (70 mM KCl, 30 mM HEPES, 5 mM MgCl2, 3 mM EGTA, at pH 7.4 with KOH) and unroofing was performed with a squirt of 2% PFA in stabilization buffer (EM grade, freshly prepared, Electron Microscopy Science #15710) on the cells using a 21-gauge needle and syringe. Afterwards, the unroofed cells were placed in fresh 4% PFA for 15 min at 21 °C and then used for immunostaining.

## Immunostaining of PREM samples

After fixation the coverslips were washed in stabilization buffer once and fixation was quenched with 50 mM NH4Cl in stabilization buffer for 7 min and washed two more times. Cells were blocked for 1 h with 4% (v/v) horse serum and 1% (w/v) bovine serum albumin (BSA) in stabilization buffer. The samples were then incubated with anti-clathrin heavy chain (P1663) antibody (1:100 dilution, #2410, Cell Signaling Technology) and 1% BSA in stabilization buffer at 21 °C for 1 h followed by 4 washing steps with 1% BSA in stabilization buffer. Next, cells were incubated with goat anti-rabbit IgG Alexa Fluor 568 (1:500 dilution, #A-11011, Invitrogen) and CellMask Deep Red Plasma Membrane Stain (1:5000, # C10046, Invitrogen) with 1% BSA in stabilization buffer for 45 min. Samples were rinsed 4 times with stabilization buffer, postfixed in 4% (w/v) paraformaldehyde in stabilization buffer for 10 min and quenched as described above prior to imaging by spinning disc confocal microscopy.

## Platinum replica preparation

After spinning disc confocal microscopy, the plasma membrane sheets were fixed in 2% glutaraldehyde in stabilization buffer for at least 30 min and EM samples were prepared as described previously[29,30]. Samples were rinsed 3 times with water and stained with 0.1% (w/v) tannic acid for 20 min followed by staining with 0.1% (w/v) uranyl acetate for 20 min. The coverslips were then dehydrated through a series of increasing ethanol concentration to 100% ethanol followed by critical point drying (Leica EM CPD300). The coverslips were then low angle rotary shadowed with 1.4 nm platinum and 5 nm carbon in a dual ion beam evaporator (Leica EM ACE600).

## Electron microscopy

Platinum and carbon coated coverslips were mounted with double sided carbon disks and imaged at a Helios 5CX scanning electron microscope. Low resolution scans for navigation were done with ETD or ICE detectors using MAPS software. Alignment of fluorescence microscopic overview images with SEM tile sets to navigate to cells of interest was done with MAPS as well. Ultrahigh resolution scanning of unroofed cells was done with TLD detector in secondary electron mode at 3.7 mm working distance, 5 kV, 21 pA, 1 μs dwell time, line integration mode (8 cycles) and 0.67 nm pixel size. Tile sets were stitched with the Grid/Collection stitching plugin in ImageJ. Images were $2 \times 2$ binned.

## Correlative light electron microscopy for the PREM samples

Correlation of fluorescence microscopic and SEM images was achieved by taking overview images of the CellMask signal using 10× magnification for navigation and by marking the region on the coverslip used for fluorescence microscopic imaging with a diamond pen. After SEM imaging the fluorescence microscopic images were aligned to the ultrahigh resolution SEM images. Coarse alignment was done based on the CellMask staining and the cell borders, exact alignment was done based on the clathrin staining using the BigWarp plugin in ImageJ.

## Spinning disc confocal microscopy

Fluorescence images were acquired on an inverted IX71 microscope (Olympus) equipped with a CSU-X1 spinning disk unit (Yokogawa) and an iLas laser illumination system (Gataca Systems) with 491 nm, 561 nm and 639 nm lasers for illumination. 10× NA 0.3 air or 60× NA 1.42 oil objectives (Olympus) were used, and images were captured with an ORCA Flash 4.0LT sCMOS camera (Hamamatsu). The system was operated using the software MetaMorph.

## Image analysis and quantification of colocalization

Image analysis and fluorescence intensity quantification were performed with ImageJ[75]. The percentage of colocalization between the two channels (Organelle and Virus/GEM) imaged was quantitatively determined on a per-object basis using a custom-made pipeline in CellProfiler[76]. In brief, the Z-stacks acquired for each channel were first split into individual images that were then segmented into objects identifying either the VLPs/GEMs or the specific organelles inside cells. The percentage of colocalization was then calculated as the amount of overlapping pixels between the identified objects in the two channels divided by the total pixel area occupied by the Virus/GEM channel.

## Calculation of energy gain for joint particle wrapping in tubules

We numerically determined the energy gain for the joint wrapping of GEM-GFP particles in tubules by minimizing the sum of bending and adhesion energies for the rotationally symmetric shapes of the membrane tubules and for membrane segments wrapping single particles as previously described[32]. For the energy minimization, the profiles of the rationally symmetric membranes around the particles are discretized into up to about 400 segments in the parametrizations as previously described[32]. To avoid membrane overlap in nearly closed membrane necks obtained for large values of the adhesion potential depth U (small values of $K_D$), the distance of membrane midplanes in these necks is constrained to be larger than 5 nm. The minimum-energy shapes were determined via constrained minimization with the FindMinimum function of the program Mathematica 13 [Wolfram Research, Inc., Mathematica, Version 13.2, Champaign, IL (2022)].

## Reporting summary

Further information on research design is available in the Nature Portfolio Reporting Summary linked to this article.

## Data availability

The microscopy data shown in the main figures is deposited at https://github.com/AG-Ewers/GEM-project. Raw data supporting the findings of this manuscript are too large to be deposited and are available from the corresponding author upon reasonable request. A reporting summary for this Article is available as a Supplementary Information file. Source data are provided with this paper.

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

## Acknowledgements

We thank Alexia de Caro, Helen Wildenauer, Gabriel Huerta Lopez and Aurora Elhazaz Fernandez for testing reagents and performing initial test experiments. We thank Dr. Amin Zehtabian for initial optimization of analysis pipelines. We thank Dr. Giulia Glorani, Max Ruwolt and Nicole Dimos for providing bacterial expression reagents. The work was funded by Deutsche Forschungsgemeinschaft (DFG) through ViroCarb - FOR2327, SFB 756, and project no. 278001972 - TRR 186. J.W. Taraska is supported by the Intramural Research Program of the National Heart, Lung, and Blood Institute, National Institutes of Health, Bethesda, Maryland, USA. We would like to acknowledge the assistance of Dr. Kai Ludwig from the Core Facility BioSupraMol supported by the DFG. We thank the microscopy facility of the FMP for technical support and Martin Lehmann for cooperation with microscopy facility of the FMP. We thank Veronica Falconieri Hays and Falconieri Visuals LLC for the illustrations. We acknowledge technical support from Andrea Senge and Carolin Knappe.

## Author contributions

H.E. and R.G. conceived and designed research; R.G., K.V.S., P.M.M., P.R. and C.S.L performed experiments and analyzed data; R.G. and U.N. developed and optimized protein purification assays; D.P., C.M. and J.T. contributed to PREM experiments; R.D. provided new reagents and analytical tools; T.W. performed the theoretical modeling; R.G., T.W. and H.E. wrote the manuscript with input from all authors.

## Funding

## Competing interests

The authors declare no competing interests.
