## [Peer Review File · Nature Communications]

Adhesion energy controls lipid binding-mediated endocytosisREVIEWER COMMENTS

Reviewer #1 (Remarks to the Author):

This is a nice paper and I enjoyed reading it. It has the potential to make valuable contributions to the field of endocytosis. However, I have few questions and comments that I would like to be addressed before I can fully appreciate all the claims made in the paper:

1. Do we have an experimental evidence of multiples nanoparticles inside a tubule? Is Fig. 2G an example of a nanoparticle contained in a tubule?
2. What is the typical size of tubules observed in the experiments?
3. Fig. 5D (ii) does not show the catenoid regions, which reduce the bending energy. These regions are very clear in Figs. 1, 5 and 7 of ref 32.
4. Authors say that in intact cells, GEMs vesiculate quickly after membrane deformation. Does it indicate a geometric instability? The notion of discontinuous transition is discussed in Phys. Rev. E 69, 031903 and Soft Matter, 2017, 13, 1455 (adhesion energy is analogous to polymerization energy). Can authors comment on this?
5. Could intact cells and energy-depleted cells have different membrane tension? Could this also contribute to the observation that tubules are not observed in intact cells but are observed in energy-depleted cells? Could this be relevant-Biophys J. 2017 Oct 17;113(8):1768-1781? Also, Soft Matter, 2017, 13, 1455 shows that higher tension promotes Omega-shaped vesicles, which in turn can promote tubulation based on the catenoid-mechanism.

Reviewer #2 (Remarks to the Author):

The paper by Groza et al contributes to the field of particle (understood in a broad sense, whether they are particles or toxins or pathogens) endocytosis and the mechanical deformation of the cell membrane. Authors have found a correlation linking membrane deformation and adhesion energy between the host and the particle/pathogen which if high enough seems to be sufficient to overcome the stiffness of the

cell membrane. The idea that the strength of adhesion might be the driving force to overcome mechanical curvature of membranes is appealing and the paper has made good progress to experimentally demonstrate this correlation using synthetic systems and cells. The first part of the paper is strong, i.e. that the energy of adhesion controls endocytosis and that this happens in a clathrin-independent manner. However, I feel that the correlation between adhesion energy and mechanical deformation has not been established neither experimentally nor through the model that they developed to understand the experimental data.

'Energy depletion' of cells is used through the manuscript and yet I am not sure what authors intend to demonstrate by using these conditions in these experiments. For example Fig 1 B shows membrane deformation for 'energy depleted cells' but no data is shown for 'standard cells' (i.e. in non energy-depleted conditions).

Figure 2A shows a schematic representation of the synthetic system. I understand that GEM are 40 nm particles with GFP as membrane proteins. How the GPI-anchor is incorporated into the membrane is not clear.

Figures 1-3 contain important phenomenological data that show how binding of GEMs to membranes lead to invaginations and endocytosis that is cholesterol and clathrin independent.

Figure 4 is key to the novel concept of the paper and it shows that the strength of the ligand-receptor interaction is related to endocytosis and membrane deformation. As before, cells are used in energy-depleted conditions, and it'd be good to know what happens with control cells. Importantly, authors demonstrate that the process of mechanical deformation of the membrane seems to be triggered by an 'affinity threshold'. This phenomenological observation is interesting and the core aspect of the paper and the one which deserves more attention – if confirmed and understood can reveal a fundamental property of particle endocytosis and has the potential to open up new research and applications. Other receptors are modulated by physical properties following this 'threshold' pattern. For example, it has been reported that there is a threshold of stiffness of 5 kPa in extracellular matrix that activates the actin-integrin-ECM molecular clutch. It is unavoidable to consider whether there are similarities between these two processes, i.e. activation of membrane related events by a threshold of. It is less clear what's the molecular mechanisms by which once the threshold in the adhesion-energy is met, the deformation of the membrane occurs.

The potential role of actin is mentioned in the discussion but no experiments are done to investigate any actin-mediated effect which can be important. The role of actin might have been underappreciated due to the energy-depleted conditions used throughout the manuscript.

The effort to add some theoretical modelling into the paper is valuable. The model that authors have used is phenomenological and based on the use of an active potential that scales with the inverse of the contact area between ligands. I don't think that this is enough to theoretically account for membrane deformation. I find the model qualitative and more a way of thinking about the experimental data than a quantitative theoretical framework that can be used to make a comparison with the experimental data. The importance of the model in the paper should be tone down.

Minor

Line 99 mentions Figure 1E which can't be found

Reviewer #3 (Remarks to the Author):

This article addresses the fundamental question of what are the mechanisms mediating internalization of lipid-binding small particles such as virions via plasma membrane deformation and endocytosis. Authors propose a smart bottom-up approach to control adhesion energy via the use of GEM nanoparticles and anti-GFP nanobodies anchored to GPI proteins, and characterize membrane deformation and particle internalization with different methods (optical and electron microscopy, FACS, modelling..)

This work provides an original perspective on a complex process with several intriguing approaches and results of a broad interest for the cell biology community. Data are well-supported and spread out to very different directions, despite lacking some in-depth study from time to time and further detail, as indicated in the following paragraph. The discussion is a bit long but very interesting and allows to zoom-out on several open questions that could be addressed with this novel approach. To my opinion, I recommend publication of this work on Nature Communications after having answered to these concerns and providing a revised version of the manuscript.

Major points:

1) Despite the interest of comparing 3 different lipid-binding viruses of similar size, the data related to induced membrane deformation and endocytosis remain quite superficial. In particular, I am not totally convinced that this involves non-clathrin-mediated endocytosis in the three cases. Is it well-known from the literature? For the 3 viruses? What does this dataset add to the current knowledge? If this is an important assumption to justify the following part, why not verifying the presence of transferrin in virus-positive endosomes, and their uptake in the presence of inhibitors of clathrin-coated pits? Panels A/C/D of Figure 1 are well built and robust, however I couldn't really interpret panel B regarding membrane deformation (show control images? Add arrows?)

2) Figure 2 is very interesting but raises some questions on the fate of GEM particles in this artificial cell assay. Concerning panel 2B and movie 1, is it acquire hiw long after GEMs addition? Is the observed dynamics similar to what is known as classical GPI proteins within the plasma membrane? Panel 2C: as in the previous point , not very clear to me; Panles 2D/E: Need quantification (especially given that together with Panels G/H the claim is quite strong, and no perturbation experiments are provided as a support)

3) Figure 3 is technically very impressive and interesting, however main conclusions come from Panel 3E only, that is very dense and complex (also, I would advice to validate the FACS approach with another method (optical microscopy for example) at least in one condition

4) Figure 4 includes the most interesting set of data and has potential applications in the future of very broad interest. Several control experiments are provided in the supplementary informations. I just have a few questions: is it possible to quantify binding more precisely in all conditions? As before, I don't find very clear Panel 4C related to membrane deformation in cells. Again, panel 4E is very dense and doesn't add much to the conclusions of Panel 3E (or am I missing something?)

5) Figure 5 concerns a theoretical approach to investigate further how adhesion energy impacts membrane deformation. Despite the general idea is interesting, I found this part not well integrated with the rest (the hypothesis and question you want to address is not clearly stated, and the conclusions of this paragraph are relatively obscure). I think the text needs to be revised to highlight the 2 regimes (wrapping and not-wrapping) and the parallels with experimental data.

We thank the reviewers for their productive input and their helpful suggestions. Please find below our answers to all points raised by the reviewers. We added additional experimental data in several Figures in the main text and supplementary data. We furthermore made changes in the text to address all points raised by the reviewers. We added page numbers to help the reviewers navigate through the manuscript updates. We also emphasized all changes in the manuscript text in yellow for easy access. We hope our manuscript in its improved form is found to be acceptable for publication by the reviewers.

Reviewer #1 (Remarks to the Author):

This is a nice paper and I enjoyed reading it. It has the potential to make valuable contributions to the field of endocytosis. However, I have few questions and comments that I would like to be addressed before I can fully appreciate all the claims made in the paper:

1. Do we have an experimental evidence of multiples nanoparticles inside a tubule? Is Fig. 2G an example of a nanoparticle contained in a tubule?

We thank the reviewer for this important question. The fluorescence micrographs in Fig. 2C and D show continuous tubules emanating from membranes that contain multiple individually resolved fluorescent particles or continuous GEM staining that must emanate from multiple particles. Furthermore, we do indeed have reason to believe that Fig. 2G shows several nanoparticles in a tubule. The scalebar is 50 nm and the green correlative fluorescence is clearly elongated and thus not resulting from a single diffraction-limited spot. It must thus be the result of several particles located in an elongated arrangement like in a tubule. We express this explicitly in the text, on page 6, lines 27-29.

2. What is the typical size of tubules observed in the experiments?

In the GPMVs and cells, confocal fluorescence micrographs showed tubules as diffraction limited lines, suggesting that they were below 200 nm in diameter. In our platinum-replica electron microscopy images, the tubes were typically below 100 nm in diameter (see Fig. 2G, second from left, 2H, rightmost). So we assume that they form tight fitting tubules around GEMs.

3. Fig. 5D (ii) does not show the catenoid regions, which reduce the bending energy. These regions are very clear in Figs. 1, 5 and 7 of ref 32.

This observation of the reviewer is correct. Figure 5D is not direct evidence, but a conceptual illustration of the particles in a tubule and we try to convey the difference between the wrapped single particle and the tubule situations here. We do not have experimental evidence besides Fig. 2G as to the shape of tubules in cells, though elongated tubules exhibit catenoid regions according to our model based on membrane biophysics. We show here a simplified model of the tubule. To avoid misinterpretations, we now clarify in the figure legend to Figure 5 that the illustration represents a case close to the wrapping threshold, where particles are only weakly wrapped.

4. Authors say that in intact cells, GEMs vesiculate quickly after membrane deformation. Does it indicate a geometric instability? The notion of discontinuous transition is discussed in Phys. Rev. E 69, 031903 and Soft Matter, 2017, 13, 1455 (adhesion energy is analogous to polymerization energy). Can authors comment on this?

We thank the reviewer for this insightful comment. Vesiculation in intact cells is likely due to active scission processes as we now address more specifically in the main text. Absence of such scission processes is a kinetic modeling requirement.

Tubulation requires a continuous wrapping transition, i.e. it requires partially wrapped states. In the model, the partially wrapped states occur due to the finite range of the adhesion potential. In, Phys. Rev. E 69, 031903, partially wrapped states are stabilised by membrane tension for a contact adhesion potential (with potential range 0). Overall, we don't see that tension can lead to scission in particle wrapping.

5. Could intact cells and energy-depleted cells have different membrane tension? Could this also contribute to the observation that tubules are not observed in intact cells but are observed in energy-depleted cells? Could this be relevant-Biophys J. 2017 Oct 17;113(8):1768-1781? Also, Soft Matter, 2017, 13, 1455 shows that higher tension promotes Omega-shaped vesicles, which in turn can promote tubulation based on the catenoid-mechanism.

We thank the reviewer for the insightful comment and are aware of the excellent work mentioned by the reviewer. Yes, they could indeed have different membrane tension. But we don't see that this contributes to having no tubules in intact cells. In general, tension increases the threshold adhesion energy required for wrapping, for sufficiently large tension values σ for which the crossover length $\sqrt{\kappa/\sigma}$ is of the size of or smaller than the particle radius. However, tension rather stabilises partially wrapped states, which can lead to, and are required for, tubulation.

Reviewer #2 (Remarks to the Author):

The paper by Groza et al contributes to the field of particle (understood in a broad sense, whether they are particles or toxins or pathogens) endocytosis and the mechanical deformation of the cell membrane. Authors have found a correlation linking membrane deformation and adhesion energy between the host and the particle/pathogen which if high enough seems to be sufficient to overcome the stiffness of the cell membrane. The idea that the strength of adhesion might be the driving force to overcome mechanical curvature of membranes is appealing and the paper has made good progress to experimentally demonstrate this correlation using synthetic systems and cells.

The first part of the paper is strong, i.e. that the energy of adhesion controls endocytosis and that this happens in a clathrin-independent manner. However, I feel that the correlation between adhesion energy and mechanical deformation has not been established neither experimentally nor through the model that they developed to understand the experimental data. 'Energy depletion' of cells is used through the manuscript and yet I am not sure what authors intend to demonstrate by using these conditions in these experiments. For example Fig 1 B shows membrane deformation for 'energy depleted cells' but no data is shown for 'standard cells' (i.e. in non energy-depleted conditions).

We thank the reviewer for this important question and apologize that we did not make the use of this approach sufficiently clear. Many lipid-dependent endocytic processes are quite fast and we observed that particles that are capable of membrane deformation *in vitro* become internalized in cells too quickly to allow for the capture of elongated, tubular membrane invaginations simply because the cell recognizes the curvature induced by the single particle early on and internalizes the particle. The depletion of cellular energy is thus a well-established paradigm to allow the observation of membrane deforming capabilities in the absence of (energy-dependent) membrane scission as it prohibits the cell to perform mechanical work to internalize particles¹⁻³. We now incorporate explicit information on the matter on page 4, lines 7-9. Furthermore, we show early endocytic events in cells that were not depleted of energy in Figure 2G. These represent the invaginations formed in cells in their native state.

Figure 2A shows a schematic representation of the synthetic system. I understand that GEM are 40 nm particles with GFP as membrane proteins. How the GPI-anchor is incorporated into the membrane is not clear.

We thank the reviewer for this important question and apologize that we did not make this sufficiently clear. Indeed, GEMS are globular nanoparticles assembled from 180 fusion proteins that contain GFP and thus expose 180 GFP molecules on their surface. These are used in our study as cell ligands. The cellular receptors to this GFP-studded capsid are nanobodies coupled to GPI-anchors in the plasma membrane. The GPI-anchor is synthesized in the cells after transfection. Cells express fusion constructs of anti-GFP nanobodies with a sequence that leads to the attachment to a GPI-anchor in the Golgi apparatus. These proteins are synthesized in the cell and delivered to the plasma membrane upon transient expression, where they expose the nanobody on the surface of the cell. We now explain this in the text, on page 4, line 26.

Figures 1-3 contain important phenomenological data that show how binding of GEMs to membranes lead to invaginations and endocytosis that is cholesterol and clathrin independent. Figure 4 is key to the novel concept of the paper and it shows that the strength of the ligand-receptor interaction is related to endocytosis and membrane deformation. As before, cells are used in energy-depleted conditions, and it'd be good to know what happens with control cells.

We thank the reviewer for this important question. As mentioned above, in control cells, particles become endocytosed so quickly that tubulation as a readout of membrane deformation cannot be observed. Furthermore, we show early endocytic events in cells that are not depleted of energy in Figure 2G, which would thus represent the invaginations as formed in control cells. The data in 4D and E on endocytosis are also from non-energy-depleted cells. Technically, we cannot measure endocytosis in energy-depleted cells as the fission of the GEM-containing tubules requires energy and GEMs technically remain surface connected in the tubules.

Importantly, authors demonstrate that the process of mechanical deformation of the membrane seems to be triggered by an 'affinity threshold'. This phenomenological observation is interesting and the core aspect of the paper and the one which deserves more attention – if confirmed and understood can reveal a fundamental property of particle endocytosis and has the potential to open up new research and applications. Other receptors are modulated by physical properties following this 'threshold' pattern. For example, it has been reported that there is a threshold of stiffness of 5 kPa in extracellular matrix that activates the actin-integrin-ECM molecular clutch. It is unavoidable to consider whether there are similarities between these two processes, i.e. activation of membrane related events by a threshold of . It is less clear what's the molecular mechanisms by which once the threshold in the adhesion-energy is met, the deformation of the membrane occurs.

We thank the reviewer for the thoughtful appreciation of our work and the exciting example. Indeed, mechanisms triggered by bimodal transitions seem to occur at many points in cells and we now specifically mention the famous molecular clutch in the manuscript. We are not a mechanobiology laboratory but are aware of the molecular clutch mechanism proposed by Mitchison and Kirschner, found by Sheetz et al and quantified

beautifully by Elosegui-Artola et al. which we now mention in the text, on page 13, lines 35-38, and many others over the last decades. At the time, we do not have any evidence pointing to a mechanosensitive protein involved in the endocytic process here, but we will include this thought in future analysis.

We agree that in the future, the specific molecules involved in abscission and in downstream processing of early endocytic vesicles must be identified to further elucidate the elusive pathways of clathrin-independent endocytosis. We envision the molecular mechanism of membrane deformation as follows: A multivalent globular particle, a ball with many binding sites distributed over its surface, attaches to a first receptor in the cell membrane. Since it has many binding sites and many receptors are available in the target membrane, due to the shape of the globular particle, every additional binding event must lead to a deformation of the membrane. In the process of this, the particle will become engulfed in membrane. The energetic cost required to deform the membrane in this way stems from the adhesion energy a.k.a. the energy released from the individual binding interactions. From our observations it seems that only once a threshold in adhesion energy is met, the membrane can be deformed in this way. As the particles become endocytosed in dependence of this process and endocytosis remains energy dependent, cellular factors must be involved in this process. In future work, we aim to identify the factors involved in this process.

The potential role of actin is mentioned in the discussion but no experiments are done to investigate any actin-mediated effect which can be important. The role of actin might have been underappreciated due to the energy-depleted conditions used throughout the manuscript.

We thank the reviewer for this important question. Actin has indeed been reported to be essential for membrane scission in many lipid-dependent endocytosis assays. We now performed additional experiments and found that pharmacological interference with actin through CytochalasinD detectably reduces the internalization of GEMs in the FACS assay, but less than for the SV40 virus (see Figure 3E). However, as expected, it had no detectable effect on Transferrin endocytosis (not included in the Figure). We now discuss this in the main text on page 8, lines 30-34.

The effort to add some theoretical modelling into the paper is valuable. The model that authors have used is phenomenological and based on the use of an active potential that scales with the inverse of the contact area between ligands. I don't think that this is enough to theoretically account for membrane deformation. I find the model qualitative and more a way of thinking about the experimental data than a quantitative theoretical framework that can be used to make a comparison with the experimental data. The importance of the model in the paper should be tone down.

We thank the reviewer for pointing this out. As our system provides a possibility to manipulate a single biophysical parameter in a complex process like endocytosis, we found it would make a lot of sense to incorporate this into a standard model to give the cell biology reader the opportunity to conceptualize the consequences of our observations. We did not mean to overemphasize the model and now changed this in the text. We now removed the separate heading for the model to tone down its importance and made several more edits to clarify the role of our model.

The main aims of the modeling are:

- to relate the dissociation constant K_d of the experiments to adhesion energies
- to calculate and compare the energies for cooperative wrapping and individual. The clear energy gain of cooperative wrapping is a quantitative aspect of the modeling, and an energetic requirement for tubulation.

However, the model is also qualitative in the sense that the threshold adhesion energy required for wrapping (or threshold value of K_d for wrapping) cannot be "predicted". Adjusting the adhesion threshold in the model to the experimental observation requires fitting a parameter, which is then used in the calculation of the energy gain for tubulation.

We modified the text to clarify these aspects, but also to shorten and tone down the modelling part, on pages 11-12.

Minor

Line 99 mentions Figure 1E which can't be found

We thank the reviewer for pointing this out. This was a mistake in writing, we removed the mention of Figure 1E.

Reviewer #3 (Remarks to the Author):

This article addresses the fundamental question of what are the mechanisms mediating internalization of lipid-binding small particles such as virions via plasma membrane deformation and endocytosis. Authors propose a

smart bottom-up approach to control adhesion energy via the use of GEM nanoparticles and anti-GFP nanobodies anchored to GPI proteins, and characterize membrane deformation and particle internalization with different methods (optical and electron microscopy, FACS, modelling..)

This work provides an original perspective on a complex process with several intriguing approaches and results of a broad interest for the cell biology community. Data are well-supported and spread out to very different directions, despite lacking some in-depth study from time to time and further detail, as indicated in the following paragraph. The discussion is a bit long but very interesting and allows to zoom-out on several open questions that could be addressed with this novel approach. To my opinion, I recommend publication of this work on Nature Communications after having answered to these concerns and providing a revised version of the manuscript.

We thank the reviewer for this assessment of our work.

Major points:

1) Despite the interest of comparing 3 different lipid-binding viruses of similar size, the data related to induced membrane deformation and endocytosis remain quite superficial. In particular, I am not totally convinced that this involves non-clathrin-mediated endocytosis in the three cases. Is it well-known from the literature? For the 3 viruses? What does this dataset add to the current knowledge? If this is an important assumption to justify the following part, why not verifying the presence of transferrin in virus-positive endosomes, and their uptake in the presence of inhibitors of clathrin-coated pits? Panels A/C/D of Figure 1 are well built and robust, however I couldn't really interpret panel B regarding membrane deformation (show control images? Add arrows?)

We thank the reviewer for pointing this out. We here used a family of viruses that are known to bind to glycolipids as test case for lipid binding-mediated membrane deformation as a common property. Indeed, while all are reported to bind to glycolipids they seem to exhibit somewhat promiscuous endocytic preferences, however, at least for the JC virus, endocytosis has not been studied at depth. We intentionally included viruses with a spectrum of endocytic pathways to emphasize their common biophysical properties: i) lipid binding and ii) the capability to deform membranes when the cellular machinery is not active. We wanted to convey here a purely biophysical point of view of viruses as globular lipid binders, oblivious to downstream cell biology of infection. We now added arrows to emphasize the tubular invaginations created by the virions in panel B. To address the reviewers concerns, we now show that all viruses become internalized in our cells and transported to late endosome/Lysosomes (see Figure 1 C,D).

2) Figure 2 is very interesting but raises some questions on the fate of GEM particles in this artificial cell assay. Concerning panel 2B and movie 1, is it acquire how long after GEMs addition? Is the observed dynamics similar to what is known as classical GPI proteins within the plasma membrane?

The image of GEMs binding to a cell and the movie demonstrating their mobility on the cell were acquired right after binding. We did not specifically quantify movement at this point, but it seems slower than that of individual lipid-anchored proteins compared to single molecule diffusion measurements we have performed before⁴⁻⁶. Indeed we have shown before that lipid-binding viruses slow down significantly when bound to their lipidic receptor in cellular and even artificial membranes due to multivalent binding^{7,8,9}. We agree with the reviewer that it would be interesting to investigate the multivalent binding of GEMs to GPI- anchored proteins from a membrane dynamics perspective. In a future project we are preparing for such investigations that would also need to include measurements in artificial membranes.

Panel 2C: as in the previous point, not very clear to me; Panels 2D/E: Need quantification (especially given that together with Panels G/H the claim is quite strong, and no perturbation experiments are provided as a support)

We thank the reviewer for pointing this out, indeed we do have quantified the colocalization of surface-bound GEMs with clathrin and caveolin, these data are in Figure S7. They are only mentioned at a later point when all different NB affinities are compared. We now included a callout to Figure S7 earlier. We also added arrows to Panel D to emphasize membrane tubules.

3) Figure 3 is technically very impressive and interesting, however main conclusions come from Panel 3E only, that is very dense and complex (also, I would advice to validate the FACS approach with another method (optical microscopy for example) at least in one condition.

We thank the reviewer for this assessment. As GEMs are new to being used as endocytic probes, we intended to document their cell biology in detail. We are grateful the reviewer appreciates our efforts. We now added fluorescence light microscopy assays of GEM endocytosis in new Figure S8.

4) Figure 4 includes the most interesting set of data and has potential applications in the future of very broad interest. Several control experiments are provided in the supplementary informations. I just have a few questions: is it possible to quantify binding more precisely in all conditions? As before, I don't find very clear Panel 4C related to membrane deformation in cells. Again, panel 4E is very dense and doesn't add much to the conclusions of Panel 3E (or am I missing something?)

We thank the reviewer for this question. We present here binding assays based on GEM fluorescence, which we feel is a reasonable proxy of GEM number attached to cells in combination with a state of the art sCMOS camera with linear detection (Figure S5). As the GPI-anchored nanobodies are recombinantly expressed in cells, we cannot perform QCM-D or Biacore measurements at the time. We are preparing to synthesize GPI-nanobodies for in vitro measurements of binding with supported membrane bilayers, but this is a new PhD-project together with chemist collaborators. We however did clearly show that co-expression of RFP with GPI-nanobodies of all affinities was constant across our binding assays, suggesting an equal level of expression for nanobodies of all affinities (Figure S6). To make panel 4C more accessible, we now added arrows to emphasize GEM-containing membrane tubules in Cells. We agree with the reviewer that Panel 4E does not change the conclusions derived from panel 3E, however it does add the information that the binding affinity of nanobodies does not influence the endocytic pathway, which we believe is important to know and impacts conclusions about the pathway.

5) Figure 5 concerns a theoretical approach to investigate further how adhesion energy impacts membrane deformation. Despite the general idea is interesting, I found this part not well integrated with the rest (the hypothesis and question you want to address is not clearly stated, and the conclusions of this paragraph are relatively obscure). I think the text needs to be revised to highlight the 2 regimes (wrapping and not-wrapping) and the parallels with experimental data.

We apologize to the reviewer for not being sufficiently clear and attempted to rewrite the text according to the reviewer's suggestions, on page 12, lines 2-19.

References

1. Ewers, H., Römer, W., Smith, A. E., Bacia, K., Dmitrieff, S., Chai, W., Mancini, R., Kartenbeck, J., Chambon, V., Berland, L., Oppenheim, A., Schwarzmann, G., Feizi, T., Schwille, P., Sens, P., Helenius, A. & Johannes, L. GM1 structure determines SV40-induced membrane invagination and infection. *Nat Cell Biol* 12, 11–8; sup pp 1-12 (2010).
2. Römer, W., Berland, L., Chambon, V., Gaus, K., Windschiegl, B., Tenza, D., Aly, M. R. E., Fraisier, V., Florent, J.-C., Perrais, D., Lamaze, C., Raposo, G., Steinem, C., Sens, P., Bassereau, P. & Johannes, L. Shiga toxin induces tubular membrane invaginations for its uptake into cells. *Nature* 450, 670–675 (2007).
3. Römer, W., Pontani, L.-L., Sorre, B., Rentero, C., Berland, L., Chambon, V., Lamaze, C., Bassereau, P., Sykes, C., Gaus, K. & Johannes, L. Actin dynamics drive membrane reorganization and scission in clathrin-independent endocytosis. *Cell* 140, 540–553 (2010).
4. Albrecht, D., Winterflood, C. M. & Ewers, H. Dual color single particle tracking via nanobodies. *Methods and Applications in Fluorescence* 1–8 (2015) doi:10.1088/2050-6120/3/2/024001.
5. Albrecht, D., Winterflood, C. M., Sadeghi, M., Tschager, T., Noé, F. & Ewers, H. Nanoscopic compartmentalization of membrane protein motion at the axon initial segment. *J Cell Biol* 215, 37–46 (2016).
6. Thoumine, O., Ewers, H., Heine, M., Groc, L., Frischknecht, R., Giannone, G., Poujol, C., Legros, P., Lounis, B., Cognet, L. & Choquet, D. Probing the dynamics of protein-protein interactions at neuronal contacts by optical imaging. *Chem. Rev.* 108, 1565–1587 (2008).
7. Ewers, H., Smith, A. E., Sbalzarini, I. F., Lilie, H., Koumoutsakos, P. & Helenius, A. Single-particle tracking of murine polyoma virus-like particles on live cells and artificial membranes. *Proc Natl Acad Sci USA* 102, 15110–15115 (2005).
8. Szklarczyk, O. M., González-Segredo, N., Kukura, P., Oppenheim, A., Choquet, D., Sandoghdar, V., Helenius, A., Sbalzarini, I. F. & Ewers, H. Receptor concentration and diffusivity control multivalent binding of Sv40 to membrane bilayers. *PLoS Comput. Biol.* 9, e1003310 (2013).
9. Kukura, P., Ewers, H., Müller, C., Renn, A., Helenius, A. & Sandoghdar, V. High-speed nanoscopic tracking of the position and orientation of a single virus. *Nat Methods* 6, 923–927 (2009).

REVIEWER COMMENTS

Reviewer #1 (Remarks to the Author):

I thank the authors for responding to my questions. I have the following comments and follow-up questions.

1. What is the typical tubule length seen in the experiments? Is this average length observed in experiments explained by the model?
2. I understand Fig. 5D is a conceptual illustration, but should it not demonstrate the central idea of adhesion-induced wrapping? As per the response, Figs. 5D (i) and (ii) show the geometries close to the wrapping threshold. Such a choice was not made in Ref 32. In that paper, even weak wrapping shapes exhibit waviness in the membrane shape. Would it not be better for the readers to see a more physically relevant geometry?
3. Why does the tubule in Fig. 2G have a much larger globular head? Fig. 7 in Ref 32 does not show such a geometry? What does it say about the forces acting on the tubule? An explanation can provide valuable insight to the readers.
4. The authors have not fully addressed my fourth question. Yes, scission is one part of a discontinuous transition, but geometric instability is another form of discontinuous transition that can deform an invagination prior to the scission process. That can allow early-endocytic structures to turn into mature vesicles, which could then undergo scission. There are several papers that explain this mechanism. I do not see how scission proteins can directly act on early-endocytic structures and execute membrane fission? An explanation would be appreciated here. Also, the role of tension in inducing scission was not the focus of the question, which the authors have tried to address.
5. In response to my fifth question, authors say, "But we don't see that this contributes to having no tubules in intact cells." I would appreciate any proof or argument that supports this statement. Tension can stabilize partially wrapped states. Tension can also prevent a partially wrapped state from forming. It depends on the interplay between the tension, adhesion strength, and the geometry of the particle. The analysis in Ref 32 was performed on tensionless membranes or membranes with negligible tension. It would be insightful to see a more elaborate explanation.

Overall, in my opinion, the discussion of the possible biophysical factors that could be at play could be developed further that would provide increased insights to the reader.

Reviewer #2 (Remarks to the Author):

Authors have addressed my original concerns in their rebuttal and have added new information to the revised manuscript that represent a significant improvement in regards to the original submission.

Reviewer #3 (Remarks to the Author):

The new version of the paper from Groza et al after revision has been deeply improved by the authors, following up the comments and suggestions of the reviewers and adding further data to the main manuscript. The authors thoroughly addressed all the raised concerns, providing crucial control experiments or simply improving the main text and figures in terms of clarity and respect to the pre-existing literature.

To conclude, I believe this high-quality work provides novel approaches and concepts to the understanding of endocytosis from a broad biophysical perspective, and importantly very exciting future work. I therefore recommend publication in Nature communications.

Reviewer #1 (Remarks to the Author):

I thank the authors for responding to my questions. I have the following comments and follow-up questions.

1. What is the typical tubule length seen in the experiments? Is this average length observed in experiments explained by the model?

The tubule observed in Figure 2.G has a length of 0.6 μm , the lengths of tubules observed in energy depleted cells with microscopy in Figure 4 range from 1 to 4 μm . Is it difficult to estimate an average length of tubules due to the diffraction limit of microscopy experiments.

Our theoretical modelling is focused on the energetic stability of particle-filled tubules, and does not address kinetic aspects of tubule formation or tubule length. In particular the tubule length depends on several, rather unpredictable stochastic factors such as the numbers of tubules nucleating in a certain cell membrane area, and the numbers of particles in the vicinity of tubules. This stochastic nature of tubule growth is also reflected by the range of observed tubule lengths mentioned above. Cooperative wrapping of linear chains of particles in membrane tubules was first observed in simulations by different groups about 10 years ago. This observation was unexpected, or puzzling, because standard models of particle wrapping with contact potentials (i.e., with adhesion potentials that have an effective potential range of 0) do not lead to an energetic gain for the cooperative wrapping of particles, compared to individual wrapping, and, thus, do not explain the formation of particle-filled tubules observed in the simulations. However, in all simulations in which particle-filled tubules were observed, the adhesion potential had a finite range, also for reasons of simulation efficiency. In Ref. 32, it was shown that this finite potential range can lead to a clear energetic gain for cooperative wrapping. The first step of our modeling therefore is to determine the effective adhesion potential for our experimental system, and to estimate the range of this potential. With rather rigorous elasticity calculations as in Ref. 32, we then show that cooperative wrapping in our system is indeed energetically favorable.

2. I understand Fig. 5D is a conceptual illustration, but should it not demonstrate the central idea of adhesion-induced wrapping? As per the response, Figs. 5D (i) and (ii) show the geometries close to the wrapping threshold. Such a choice was not made in Ref 32. In that paper, even weak wrapping shapes exhibit waviness in the membrane shape. Would it not be better for the readers to see a more physically relevant geometry?

The reviewer is right that the illustration in Fig. 5D is not a minimum-energy conformation as in Ref. 32. Rather, 5D depicts an intermediate state that may lead to such a minimum-energy conformation. We now clarify this in the figure caption and have also added the supplementary figure S9 with modelled minimum-energy shapes.

3. Why does the tubule in Fig. 2G have a much larger globular head? Fig. 7 in Ref 32 does not show such a geometry? What does it say about the forces acting on the tubule? An explanation can provide valuable insight to the readers.

We assume the reviewer means the third of four shown GEM-containing invaginations in Fig. 2G that shows a particular and indeed interesting shape. We agree that the observed shape is thought-inspiring, but given the absence of any additional information through markers for example, it is difficult to make statements at this time that would rise above the level of speculation. We hope the reviewer understands that we would prefer not to speculate about what exactly is happening here as we have no firm basis for this.

4. The authors have not fully addressed my fourth question. Yes, scission is one part of a discontinuous transition, but geometric instability is another form of discontinuous transition that can deform an invagination prior to the scission process. That can allow early-endocytic structures to turn into mature vesicles, which could then undergo scission. There are several papers that explain this mechanism. I do not see how scission proteins can directly act on early-endocytic structures and execute membrane fission? An explanation would be appreciated here. Also, the role of tension in inducing scission was not the focus of the question, which the authors have tried to address.

We thank the reviewer for this question. Modelling fission requires molecular models that include the lipid structure of membranes. The fission process therefore is beyond our elasticity model, and also beyond the elasticity model for clathrin-mediated endocytosis of the Agrawal group mentioned in the fourth previous question of the reviewer. We appreciate the comments and recommendations of the referee, but prefer to refrain from further speculations on fission processes in our cell systems. In general, fission is necessarily discontinuous because of the involved topology change of the membrane, and therefore associated with instabilities as a discontinuous, first-order transition.

Experiments with model membranes demonstrate that fission can occur spontaneously if membrane necks are sufficiently closed. Sufficiently large membrane tension is known to prevent necks from closing and, thus, acts against fission (see e.g. Lipowsky et al., ACS Nano 2023, 17, 11957–11968). In cell systems, fission can be assisted by proteins or biophysical mechanisms.

5. In response to my fifth question, authors say, “But we don’t see that this contributes to having no tubules in intact cells.” I would appreciate any proof or argument that supports this statement. Tension can stabilize partially wrapped states. Tension can also prevent a partially wrapped state from forming. It depends on the interplay between the tension, adhesion strength, and the geometry of the particle. The analysis in Ref 32 was performed on tensionless membranes or membranes with negligible tension. It would be insightful to see a more elaborate explanation.

We thank the reviewer for this question. For standard models with contact adhesion potentials, partially wrapped states only occur in the presence of tension. Tension leads to a discontinuous transition between states in which particles are weakly wrapped (with wrapping degree less than 50%) to states in which particles are strongly wrapped (with wrapping degree larger 50%), but does not prevent partially wrapped states per se. Rather, it enables such states in standard models with contact adhesion potentials. As mentioned above, sufficiently large membrane tension is known to prevent necks from closing and, thus, acts against fission. For adhesion potentials of finite range as in our model, partially wrapped states occur also in the absence of tension.

Overall, in my opinion, the discussion of the possible biophysical factors that could be at play could be developed further that would provide increased insights to the reader.

We thank the reviewer for this comment. We have to apologize that, as other reviewers asked us to downplay the modelling, we do not elaborate exhaustively on this topic. As emphasized in our response to point 1 above, our modelling focus is on elasticity calculations that help to understand the biophysical origin of particle-induced tubulation. We agree with the referee that kinetic aspects of tubulation, or molecularly detailed models of fission processes, or of general interest, and that a lot of potential lies in further biophysical investigation of the phenomenon we describe here in cells, which merit further investigation. We indeed plan to continue working with this system in a more biophysical setting where even more parameters can be tightly controlled. As the reviewer rightly points out, there is heterogeneity in cells and many (competing) energy-consuming and biophysical processes take place simultaneously and possibly influencing events in a not entirely homogenous or directional manner. Working with GEMs in membrane and vesicle systems in which more parameters such as membrane tension etc. are accessible will provide more insight in the future and we are preparing to do such work. For now, we try to be cautious with the interpretation of results we may not entirely understand from the biophysical viewpoint, while drawing on the broad knowledge that is available in the literature to explain our observation and to draw new conclusion from it where possible.

REVIEWERS' COMMENTS

Reviewer #1 (Remarks to the Author):

Authors have addressed my concerns. Thank you.